# Efficacy of Statin Treatment According to Baseline Renal Function in Korean Patients with Acute Myocardial Infarction Not Requiring Dialysis Undergoing Newer-Generation Drug-Eluting Stent Implantation

**DOI:** 10.3390/jcm10163504

**Published:** 2021-08-09

**Authors:** Yong Hoon Kim, Ae-Young Her, Myung Ho Jeong, Byeong-Keuk Kim, Sung-Jin Hong, Seunghwan Kim, Chul-Min Ahn, Jung-Sun Kim, Young-Guk Ko, Donghoon Choi, Myeong-Ki Hong, Yangsoo Jang

**Affiliations:** 1Division of Cardiology, Department of Internal Medicine, Kangwon National University School of Medicine, Chuncheon 24289, Korea; hermartha1@gmail.com; 2Department of Cardiology, Cardiovascular Center, Chonnam National University Hospital, Gwangju 61469, Korea; myungho@chollian.net; 3Division of Cardiology, Severance Cardiovascular Hospital, Yonsei University College of Medicine, Seoul 03722, Korea; kimbk@yuhs.ac (B.-K.K.); HONGS@yuhs.ac (S.-J.H.); DRCELLO@yuhs.ac (C.-M.A.); kjs1218@yuhs.ac (J.-S.K.); ygko@yuhs.ac (Y.-G.K.); cdhlyj@yuhs.ac (D.C.); mkhong61@yuhs.ac (M.-K.H.); jangys1212@yuhs.ac (Y.J.); 4Division of Cardiology, Inje University College of Medicine, Haeundae Paik Hospital, Busan 48108, Korea; cloudksh@gmail.com

**Keywords:** statin, myocardial infarction, renal function

## Abstract

We investigated the 2-year efficacy of statin treatment according to baseline renal function in patients with acute myocardial infarction (AMI) not requiring dialysis undergoing newer-generation drug-eluting stent (DES) implantation. A total of 18,875 AMI patients were classified into group A (statin users, *n* = 16,055) and group B (statin nonusers, *n* = 2820). According to the baseline estimated glomerular filtration rate (eGFR; ≥90, 60–89, 30–59 and <30 mL/min/1.73 m^2^), these two groups were sub-classified into groups A1, A2, A3 and A4 and groups B1, B2, B3 and B4. The major adverse cardiac events (MACE), defined as all-cause death, recurrent MI (re-MI) and any repeat revascularization, were evaluated. The MACE (group A1 vs. B1, *p* = 0.002; group A2 vs. B2, *p* = 0.007; group A3 vs. B3, *p* < 0.001; group A4 vs. B4, *p* < 0.001), all-cause death (*p* = 0.006, *p* < 0.001, *p* < 0.001, *p* < 0.001, respectively) and cardiac death (*p* = 0.004, *p* < 0.001, *p* < 0.001, *p* < 0.001, respectively) rates were significantly higher in statin nonusers than those in statin users. Despite the beneficial effects of statin treatment, the MACE (group A1 vs. A2 vs. A3 vs. A4: 5.2%, 6.4%, 10.1% and 18.5%, respectively), all-cause mortality (0.9%, 1.8%, 4.6% and 12.9%, respectively) and cardiac death (0.4%, 1.0%, 2.6% and 6.8%, respectively) rates were significantly increased as eGFR decreased in group A. These results may be related to the peculiar characteristics of chronic kidney disease, including increased vascular calcification and traditional or nontraditional cardiovascular risk factors. In the era of newer-generation DESs, although statin treatment was effective in reducing mortality, this beneficial effect was diminished in accordance with the deterioration of baseline renal function.

## 1. Introduction

During the past two decades, rapid coronary reperfusion and revascularization with newer antiplatelet and anticoagulation therapies have improved the survival of patients with acute myocardial infarction (AMI) [1,2]. Ischemic heart disease accounts for almost 1.8 million annual deaths, or 20% of all deaths in Europe [3]. Similar to Western countries, AMI continues to be a major cause of mortality in the Asia-Pacific population [4]. Statin, an inhibitor of 3-hydroxy-3-methylglutaryl-coenzyme A (HMG-CoA) reductase activity, has both fundamental lipid-lowering capacity and additional pleiotropic effects on reducing morbidity and mortality [5,6]. The current guidelines recommend that statin therapy should be initiated or continued in all patients with AMI if there are no contraindications to its use [7,8]. Every 30% decrease in glomerular filtration rate (GFR) was associated with a 29% increase in the risk of a major vascular event (MVE) [9]. Hence, individuals with chronic kidney disease (CKD) grade 3a to 4 (GFR: 15–59 mL/min/1.73 m^2^) have a 2- or 3-fold increased risk of cardiovascular mortality compared with those without CKD [10]. Some suggested mechanisms for the progression of CKD in patients with cardiovascular and renal diseases include endothelial dysfunction, oxidative stress and systemic inflammation of the glomerular capillary wall [11]. Statins alleviate many adverse effects of reduced nitric oxide availability in the inflammatory environment and improve endothelial function [12]. Moreover, statin treatment has been considered a mainstay strategy for CKD patients with respect to reducing the all-cause mortality [13]. Although previous reports [14,15,16] showed that statin treatment reduced the risk of major adverse events in patients with CKD, there are some debates [17]. Additionally, their study population [14,15,16,17] was not confined to patients with AMI. Kim et al. [18,19] showed that stent generation could be regarded as an important determinant of major adverse cardiac events (MACE) in patients with ST-segment elevation myocardial infarction (STEMI) and AMI. Therefore, we believe that the presence or absence of beneficial effects of statin treatment on major adverse events should be re-estimated in patients with AMI according to renal function under the current newer-generation drug-eluting stent (DES) era to provide more accurate real-world information to interventional cardiologists. Hence, in this study, we evaluated the 2-year efficacy of statin treatment according to baseline renal function in patients with AMI undergoing newer-generation DES implantation.

## 2. Method

### 2.1. Study Population

The study population was recruited from the Korea AMI Registry (KAMIR) [4]. Details of this registry can be found on the KAMIR website (http://www.kamir.or.kr (accessed on 15 April 2021). All patients aged ≥18 years at the time of hospital admission were included. The KAMIR was established in November 2005 and involved more than 50 communities and teaching hospitals in South Korea. A total of 45,555 patients with AMI who underwent successful stent implantation and who were not receiving continuous renal replacement therapy including hemodialysis or peritoneal dialysis between January 2006 and June 2015 were eligible for inclusion in this study. Patients with the following were also excluded: deployed bare-metal stents (*n* = 2362, 5.2%) and first-generation DES (*n* = 11,166, 24.5%), incomplete laboratory results (*n* = 8330, 18.3%), loss to follow-up (*n* = 2247, 4.9%), post-percutaneous coronary intervention (PCI) thrombolysis in myocardial infarction (TIMI) flow grade <3 (*n* = 2089, 4.6%), in-hospital death (*n* = 447, 1.0%) and treatment with other kinds of statins, except for atorvastatin, rosuvastatin, simvastatin, pitavastatin, pravastatin and fluvastatin (*n* = 39, 0.09%). Thus, a total of 18,875 AMI patients who underwent successful PCI with a newer-generation DES were included. The types of new-generation DESs used are listed in Table 1. Among the AMI patients, 16,055 (85.1%) were classified into group A (statin users) and 2820 (15.0%) into group B (statin nonusers). Thereafter, groups A and B were further subclassified into groups A1 and B1 (eGFR ≥ 90 mL/min/1.73 m^2^, *n* = 6847 (42.6%) and *n* = 889 (31.5%), respectively), groups A2 and B2 (eGFR 60–89 mL/min/1.73 m^2^, *n* = 6557 (40.8%) and *n* = 1227 (43.5%), respectively), groups A3 and B3 (eGFR 30–59 mL/min/1.73 m^2^, *n* = 2144 (13.4%) and *n* = 537 (19.0%), respectively) and groups A4 and B4 (eGFR < 30 mL/min/1.73 m^2^, *n* = 507 (3.2%) and *n* = 167 (5.9%), respectively) according to their baseline renal function and strata used to define CKD stages (Figure 1) [20]. However, because the number of patients included in stages 4 and 5 was small, they were grouped into one group (A4 or B4) in our study. The detailed reasons for not using statins in group B were as follows: (1) expected risk was higher than the benefit due to several etiologic factors such as end-stage renal failure, advanced age ≥75 years or severe heart failure (HF) (*n* = 1213, 43.0%), (2) abnormal liver function (aspartate aminotransferase or alanine aminotransferase was higher than 3-fold the upper normal limit) (*n* = 689, 24.4%), (3) multi-organ failure (*n* = 121, 4.3%), (4) statin-induced myopathy or arthralgia (*n* = 110, 3.9%) and (5) unknown (*n* = 687, 24.4%). All data were collected using a web-based case report form at each participating center. The study was conducted in accordance with the ethical guidelines of the 2004 Declaration of Helsinki and was approved by the ethics committee at each participating center and the Chonnam National University Hospital Institutional Review Board ethics committee (CNUH-2011-172). All 18,875 patients included in the study provided written informed consent prior to enrollment. They also completed a 2-year clinical follow-up through face-to-face interviews, phone calls or chart reviews. All clinical events were evaluated by an independent event adjudication committee. The event adjudication process was previously described by the KAMIR investigators. [21]. 

### 2.2. Percutaneous Coronary Intervention (PCI) Procedure and Medical Treatment

Coronary angiography and PCI were performed via a transfemoral or transradial approach in accordance with the general guidelines [22]. Aspirin (200–300 mg) and clopidogrel (300–600 mg) when available, or alternatively, ticagrelor (180 mg) or prasugrel (60 mg), were prescribed as the loading doses to the individuals before PCI. After PCI, dual antiplatelet therapy (DAPT; a combination of aspirin (100 mg/day) with clopidogrel (75 mg/day) or ticagrelor (90 mg twice a day) or prasugrel (5–10 mg/day)) was recommended for more than 12 months. Based on previous reports [23,24], triple antiplatelet therapy was administered (TAPT; 100 mg of cilostazol administered twice a day in addition to DAPT) at the discretion of the individual operator. In this study, the patients who received atorvastatin, rosuvastatin, simvastatin, pitavastatin, pravastatin and fluvastatin were included (Table 1) and the type and dose of statins to be used were left to the physicians’ discretion.

### 2.3. Study Definitions and Clinical Outcomes

AMI was defined according to the current guidelines [7,8]. A successful PCI was defined as a residual stenosis of <30% and TIMI flow grade 3 in the infarct-related artery (IRA). Glomerular function was calculated using the Chronic Kidney Disease Epidemiology Collaboration (CKD-EPI) equation for eGFR [25]. The major clinical endpoint was the occurrence of MACE, defined as all-cause mortality, recurrent myocardial infarction (re-MI) or any repeat coronary revascularization, including target lesion revascularization (TLR), target vessel revascularization (TVR) and non-TVR during the follow-up period. All-cause mortality was considered cardiac death (CD) unless an undisputed noncardiac cause was present [26]. 

### 2.4. Statistical Analysis

Differences in the continuous variables among the four groups were evaluated using analysis of variance or the Jonckheere–Terpstra test, and a post hoc analysis was performed using the Hochberg test or Dunnett’s T3 test; data are presented as means ± standard deviations. For discrete variables, differences between two of the four or eight groups were analyzed using the chi-square or Fisher’s exact test, as deemed appropriate, and data are presented as counts and percentages. We tested all variables in the univariate analysis (*p* < 0.05) (Appendix A). After univariate analysis, we tested all variables with *p* < 0.001 in the multivariate Cox regression analysis, which are listed as follows: male sex, age, left ventricular ejection fraction (LVEF), body mass index (BMI), systolic blood pressure (SBP), diastolic blood pressure (DBP), cardiogenic shock, cardiopulmonary resuscitation (CPR) on admission, Killip class III/IV, STEMI, hypertension, diabetes mellitus (DM), previous MI, previous PCI, previous HF, previous cerebrovascular accident (CVA), current smoker, N-terminal pro-brain natriuretic peptide (NT-ProBNP), blood glucose level, total cholesterol level, triglyceride level, low-density lipoprotein (LDL) cholesterol level, high-density lipoprotein (HDL) cholesterol level, use of aspirin, use of angiotensin-converting enzyme inhibitor (ACEI), use of beta blockers (BBs), left main coronary artery (LM) infarct-related artery (IRA) and treated vessel, single-vessel disease, ≥3-vessel disease, stent diameter, stent length and number of stents. Various clinical outcomes were estimated using Kaplan–Meier curve analysis, and group differences were compared using the log-rank test. A two-tailed *p* value < 0.05 was considered statistically significant. All statistical analyses were performed using SPSS software v20 (IBM; Armonk, NY, USA).

## 3. Results

### 3.1. Baseline Characteristics

Table 1, Table 2, Table 3 and Table 4 show the baseline, laboratory, angiographic and procedural characteristics of the study population.

#### 3.1.1. Group A (Statin Users)

Group A1 (eGFR ≥ 90 mL/min/1.73 m^2^) included the highest number of male patients; patients who received PCI within 24 h; current smokers; those with left anterior descending coronary artery (LAD) and left circumflex coronary artery (LCx) as the IRA and treated vessels, American College of Cardiology/American Heart Association (ACC/AHA) type B2 lesion, single-vessel disease and biolimus-eluting stent (BES) as the deployed stent; those who used optical coherence tomography and fraction flow reserve; and those prescribed with aspirin, ticagrelor, prasugrel, ACEI, BB and rosuvastatin as the discharge medications. The mean levels of LVEF, BMI, SBP, DBP, total cholesterol, triglyceride and LDL cholesterol and the mean diameter of deployed stents were highest in group A1. In group A2 (eGFR 60–89 mL/min/1.73 m^2^), the number of patients with STEMI and pre-PCI TIMI 0/1 and the mean levels of peak creatine kinase myocardial band (CK-MB) and HDL cholesterol were highest. In group A3 (eGFR 30–59 mL/min/1.73 m^2^), including patients who required CPR on admission, who received clopidogrel and cilostazol as the discharge medications, with the right coronary artery (RCA) as the IRA and treated vessel, with 2-vessel disease and with zotarolimus-eluting stent as a deployed stent, the mean age of the enrolled patients was highest. In group A4 (eGFR < 30 mL/min/1.73 m^2^), including patients with cardiogenic shock, Killip class III/IV, non-STEMI (NSTEMI), hypertension, DM, previous MI, previous PCI, previous coronary artery bypass graft (CABG), previous HF, previous CVA, LM IRA and treated vessel, ACC/AHA type C lesion, ≥3-vessel disease, everolimus-eluting stent as a deployed stent and atorvastatin as a discharge medication, the mean values of peak troponin-I, NT-ProBNP, high-sensitivity C-reactive protein (hs-CRP), blood glucose and stent length and mean number of deployed stents were highest.

#### 3.1.2. Group B (Statin Nonusers)

Group B1 (eGFR ≥ 90 mL/min/1.73 m^2^) included the highest number of male patients, patients who received PCI within 24 h, current smokers and patients with LAD and LCx as the IRA and treated vessels, single-vessel disease, BES as a deployed stent and aspirin, prasugrel and ACEI as the discharge medications. The mean levels of LVEF, BMI, SBP, DBP, total cholesterol, triglyceride and LDL cholesterol and the mean diameter of deployed stents were highest in group B1. In group B2 (eGFR 60–89 mL/min/1.73 m^2^), including patients with STEMI, ACC/AHA type B1 lesion and pre-PCI TIMI 0/1, the mean levels of peak CK-MB and HDL cholesterol and the prescription rates of clopidogrel and BB as the discharge medications were highest. In group B3 (eGFR 30–59 mL/min/1.73 m^2^), including patients who needed CPR on admission and those with previous CABG, RCA as the IRA and treated vessel and 2-vessel disease, the mean age of enrolled patients and mean number of deployed stents were highest. In group B4 (eGFR < 30 mL/min/1.73 m^2^), including patients with cardiogenic shock, Killip class III/IV, NSTEMI, hypertension, DM, previous MI, previous PCI, previous CABG, previous HF, previous CVA, LM IRA and treated vessel, ACC/AHA type C lesion, ≥3-vessel disease, everolimus-eluting stent as a deployed stent and atorvastatin as a discharge medication, the mean values of peak troponin-I, NT-ProBNP, hs-CRP, blood glucose, stent length and mean number of deployed stents were highest.

### 3.2. Clinical Outcomes 

The 2-year major clinical outcomes are summarized in Table 5 and Appendix A and Figure 2.

#### 3.2.1. Group A

After adjustment, although the MACE (Table 5 and Figure 2A) rate was not significantly different between groups A1 and A2, it was significantly higher in groups A3 (adjusted hazard ratio (aHR), 1.465; 95% CI, 1.183–1.813; *p* < 0.001) and A4 (aHR, 2.082; 95% CI, 1.514–2.863; *p* < 0.001) than that in group A1, higher in groups A3 (aHR, 1.249; 95% CI, 1.027–1.520; *p* = 0.026) and A4 (aHR, 1.701; 95% CI, 1.263–2.290; *p* < 0.001) than that in group A2 and higher in group A4 than that in group A3 (aHR, 1.439; 95% CI, 1.059–1.954; *p* = 0.020). The all-cause death (Figure 2B) rates were significantly higher in groups A2 (aHR, 1.937; 95% CI, 1.348–2.784; *p* < 0.001), A3 (aHR, 3.691; 95% CI, 2.452–5.554; *p* < 0.001) and A4 (aHR, 5.068; 95% CI, 3.037–8.459; *p* < 0.001) than that in group A1, higher in groups A3 (aHR, 1.843; 95% CI, 1.342–2.531; *p* < 0.001) and A4 (aHR, 3.160; 95% CI, 2.104–4.745; *p* < 0.001) than that in group A2 and higher in group A4 than that in group A3 (aHR, 2.060; 95% CI, 1.396–3.039; *p* < 0.001). The CD (Figure 2C) rates were significantly higher in groups A2 (aHR, 1.964; 95% CI, 1.215–3.177; *p* = 0.006), A3 (aHR, 3.429; 95% CI, 1.993–5.898; *p* < 0.001) and A4 (aHR, 4.512; 95% CI, 2.318–8.783; *p* < 0.001) than that in group A1, higher in groups A3 (aHR, 1.647; 95% CI, 1.087–2.495; *p* = 0.019) and A4 (aHR, 2.829; 95% CI, 1.674–4.781; *p* <0.001) than that in group A2 and higher in group A4 than that in group A3 (aHR, 2.040; 95% CI, 1.218–3.418; *p* = 0.007). However, the re-MI (Figure 2D) and any repeat revascularization (Figure 2E) rates were not significantly different among the four groups after adjustment (Table 5).

#### 3.2.2. Group B

Appendix A shows the HRs for the 2-year major clinical outcomes in statin nonusers. After adjustment, the rate of MACE (Figure 2A) was not significantly different between groups B1 and B2, B1 and B3 and B2 and B3. However, it was significantly higher in group B4 than those in groups B1 (aHR, 2.648; 95% CI, 1.526–4.596; *p* = 0.001), B2 (aHR, 2.055; 95% CI, 1.297–3.254; *p* = 0.002) and B3 (aHR, 1.676; 95% CI, 1.056–2.661; *p* = 0.029). The all-cause death (Figure 2B) rates were not significantly different between groups B1 and B2 and groups B2 and B3. However, they were higher in group B3 than that in group B1 (aHR, 2.014; 95% CI, 1.076–3.769; *p* = 0.029) and higher in group B4 than that in group B1 (aHR, 6.891; 95% CI, 3.114–12.25; *p* < 0.001). Moreover, the all-cause death rate was higher in group B4 than that in groups B2 (aHR, 2.914; 95% CI, 1.681–5.050; *p* < 0.001) and B3 (aHR, 2.091; 95% CI, 1.238–3.233; *p* = 0.006). Similarly, the CD (Figure 2C) rate was higher in group B3 than that in group B1 (aHR, 2.201; 95% CI, 1.054–4.596; *p* = 0.036) and higher in group B4 than that in group B1 (aHR, 8.727; 95% CI, 3.295–14.11; *p* < 0.001). Moreover, the CD rates were higher in group B4 than that in groups B2 (aHR, 2.681; 95% CI, 1.400–5.135; *p* = 0.003) and B3 (aHR, 2.022; 95% CI, 1.166–3.184; *p* = 0.014). The re-MI (Figure 2D) and any repeat revascularization (Figure 2E) rates were not significantly different among the four groups after adjustment (Appendix A).

#### 3.2.3. Group A vs. B

Table 6 shows clinical outcomes between the statin user and nonuser groups at 2 years. In the four baseline renal function groups, the rates of MACE (group A1 vs. B1, aHR, 1.573; 95% CI, 1.181–2.096; *p* = 0.002; group A2 vs. B2, aHR, 1.381; 95% CI, 1.092–1.747; *p* = 0.007; group A3 vs. B3, aHR, 1.732; 95% CI, 1.329–2.266; *p* < 0.001; and group A4 vs. B4, aHR, 1.949; 95% CI, 1.347–2.822; *p* < 0.001), all-cause death (aHR, 2.242; 95% CI, 1.261–3.984; *p* = 0.006; aHR, 2.139; 95% CI, 1.471–3.110; *p* < 0.001; aHR, 2.510; 95% CI, 1.780–3.541; *p* < 0.001; and aHR, 2.476; 95% CI, 1.629–3.755; *p* < 0.001, respectively) and CD (aHR, 2.956; 95% CI, 1.412–6.189; *p* = 0.004; aHR, 2.422; 95% CI, 1.536–3.819; *p* < 0.001; aHR, 3.150; 95% CI, 2.069–4.795; *p* < 0.001; aHR, 3.341; 95% CI, 1.975–5.706; *p* < 0.001, respectively) were higher in statin nonusers than in statin users. However, the re-MI and any repeat revascularization rates were not significantly different between the statin user and nonuser groups.

#### 3.2.4. Independent Predictors

Table 7 and Appendix A show the independent predictors for MACE and all-cause death in statin users and nonusers. Older age (≥65 years), STEMI, reduced LVEF (<40%), cardiogenic shock, CPR on admission, NT-ProBNP, LDL cholesterol, ACEI levels, ≥3-vessel disease and LM (IRA) were common independent predictors for both MACE and all-cause mortality in the statin user group (Table 7). Reduced LVEF, cardiogenic shock, CPR on admission, NT-ProBNP and LDL cholesterol levels and BB were common independent predictors for both MACE and all-cause death in the statin nonuser group (Appendix A).

## 4. Discussion

The main findings of this retrospective observational study including patients with AMI who underwent successful PCI with newer-generation DES implantation were as follows: (1) regardless of the baseline renal function, individuals who underwent statin treatment had reduced rates of MACE, all-cause mortality and CD than those in statin nonusers; (2) despite these beneficial effects of statin therapy, the MACE, all-cause death and CD rates were significantly increased as the baseline eGFR decreased; (3) older age, STEMI, reduced LVEF, cardiogenic shock, CPR on admission, NT-ProBNP, LDL cholesterol, ACEI levels, ≥3-vessel disease and LM (IRA) were common independent predictors for both MACE and all-cause mortality in the statin user group.

To date, in the current guidelines [7,8], despite an MVE reducing the benefit of statin therapy in predialysis patients, this beneficial effect of statin therapy was not distinguished according to renal function (e.g., CKD grade 3, 4, or 5), and there is no convincing evidence among patients on dialysis [10]. In our study, we only included patients with AMI who did not require dialysis. Additionally, we directly compared major clinical outcomes between the statin user and nonuser groups according to baseline renal function to evaluate the presence or absence of benefit of statin treatment in these different renal function groups. In Table 6, in all four groups (eGFR ≥ 90, 60–89, 30–59 and <30 mL/min/1.73 m^2^), statin therapy significantly reduced the rates of MACE, all-cause mortality and CD compared with those in statin nonusers. Moreover, the rates of MACE, all-cause mortality and CD significantly increased as the baseline eGFR decreased in group A (Table 5). These findings could be related to the poorer baseline characteristics of the statin nonuser group (e.g., reduced LVEF, high numbers of patients with cardiogenic shock or CPR on admission and high mean level of NT-ProBNP; Table 3 and Table 4) compared with statin users. However, our results were consistent with those of previous reports [15,27,28]. Palmer et al. [15] showed that statins reduced the all-cause mortality (relative risk (RR), 0.81; 95% CI, 0.74–1.88) and CD (RR, 0.78; 95% CI, 0.68–0.89) rates compared with placebo or no treatment in individuals not receiving dialysis. Sarnak et al. [10] also mentioned that the benefit of reducing MVE with statin-based therapy decreases as eGFR declines. Similarly, Herrington et al. [14] demonstrated that smaller relative effects of MVE were observed as eGFR declined (RR, 0.78; 99% CI, 0.75–0.82 for eGFR ≥ 60 mL/min/1.73 m^2^; RR, 0.76; 99% CI, 0.70–0.81 for eGFR 45–60 mL/min/1.73 m^2^; RR, 0.85; 99% CI, 0.75–0.96 for eGFR 30 to <45 mL/min/1.73 m^2^; RR, 0.85; 95% CI, 0.71–1.02 for eGFR <30 mL/min/1.73 m^2^). In group B, the MACE rates between groups B1 and B3 and between groups B2 and B3 and all-cause mortality and CD rates between groups B1 and B2 and between groups B2 and B3 were not significantly different (Appendix A). However, after statin treatment (group A), the rates of MACE between groups A1 and A3 (aHR, 1.465; *p* < 0.001) and groups A2 and A3 (aHR, 1.249; *p* = 0.026) and all-cause mortality and CD rates between groups A1 and A3 (aHR, 3.691; *p* < 0.001, aHR, 3.429; *p* < 0.001, respectively) and between groups A2 and A3 (aHR, 1.843; *p* < 0.001, aHR, 1.647; *p* = 0.019, respectively) were significantly different. These results could reflect the trend that if GFR is reduced, the relative beneficial effects of statins might be smaller, in accordance with previous reports. [10,14]. Although the precise mechanisms responsible for the pattern of diminished benefit of statin with lower renal function are not well-known, the peculiar characteristics of the patients with CKD could be related to this pattern [14]. Patients with CKD are often excluded from randomized trials that evaluate cardioprotective drugs, and the quality and coverage of evidence on which to guide decision making in this population is suboptimal [29]. This lack of evidence on optimal treatment strategies for such patients may result in worse outcomes [30]. Additionally, the cause of CD is influenced by misclassification of their atypical clinical presentation [31]. The difficulty of interpreting elevated levels of biomarkers of cardiac damage in CKD is a possible contributing factor [32]. As the GFR declines, vascular calcification increases, and the calcification of the intima and media of large vessels in CKD is associated with all-cause death and cardiovascular mortality [10,33,34]. These cardiovascular changes in CKD are related to traditional (e.g., diabetes and hypertension) and nontraditional CKD-related cardiovascular disease risk factors (e.g., mineral and bone disease abnormalities, inflammation and oxidative stress) [10]. Because there is geographical variation in the prevalence of DM, the absolute magnitude of beneficial effects of statin therapy can vary regionally [35].

In our study, the re-MI and any repeat revascularization rates were not significantly different between the statin user and nonuser groups. Similar results were reported by Natsuaki et al. [36]. Among 14,706 patients who underwent PCI [36], the number of patients with AMI was approximately 30%. During a median follow-up of 956 days, the re-MI and any repeat revascularization rates were not significantly different between the statin user and nonuser groups according to the three different renal function groups (eGFR ≥ 60, ≤30 to <60 and <30 mL/min/1.73 m^2^). They [36] suggested that patients with advanced CKD (eGFR < 30 mL/min/1.73 m^2^) generally have advanced atherosclerosis, typically characterized by heavy calcification, and statins may no longer provide significant benefits in patients with end-stage vascular pathology. Another randomized study [37] failed to show the effects of statin therapy in decreasing restenosis. Although the study population was not confined to individuals with AMI or CKD, according to the Cholesterol Treatment Trialists’ (CTT) Collaboration report, intensive statin therapy reduced the coronary revascularization rate by about 19% (95% CI, 11–18; *p* < 0.0001) [38]. Walter et al. [39] found that patients receiving prolonged statin treatment developed lower in-stent restenosis rates in comparison with nonreceivers (25% vs. 38%). Therefore, our results showing similar re-MI and any repeat revascularization rates between statin users and nonusers could be related to low number of enrolled patients in groups A4, B1, B3 and B4 and relatively low incidences of these events compared with previous studies [17,30]. According to recent meta-analysis data that evaluated CKD patients [40], in which CKD was defined as eGFR < 60 mL/min/1.73 m^2^, results showed that the TLR/TVR (RR, 0.69; 95% CI, 0.57–0.84) was significantly reduced with DESs compared with bare-metal stents (BMS). Additionally, the use of second-generation DESs were associated with relative 27% reduction in TLR/TVR compared with first-generation DESs. Another study’s meta-analysis data [41] showed that DESs were associated with lower TVR (RR, 0.61; 95% CI, 0.47–0.80) when compared with BMSs in patients with CKD. However, we think that future studies specifically focused on advanced CKD may help to clarify the benefit of statin treatment after PCI in this group. Interestingly, the number of patients with NSTEI was increased as their renal function deteriorated (Table 1 and Table 2). Although, the precise underlying mechanisms of this phenomenon are not well known, some suggest that plaque erosion may be more predominant in the CKD group, in those patients who tend to be older and in those who have more established atherosclerosis, whereas the incidence of plaque rupture may be more common in younger non-CKD patients in whom less mature plaques are more vulnerable to rupture [42,43]. In both the statin user and nonuser groups, reduced LVEF, cardiogenic shock, CPR on admission and NT-ProBNP and LDL cholesterol levels were common independent predictors for both MACE and all-cause mortality. These variables are well-known unfavorable risk factors for mortality in patients with AMI [7,8]. 

Because the study populations of previous studies [14,15,16,17,27,28,36] regarding the long-term effects of statin treatment on major adverse events in patients with CKD were not confined to individuals with AMI and who received newer-generation DESs, we investigated the long-term major clinical outcomes of statin therapy confined to those patients to reflect current real-world practice. Moreover, as mentioned [29], evidence on optimal treatment strategies in patients with CKD is not abundant. More than 50 high-volume universities or community hospitals in South Korea participated in this study, but the study population was insufficient to provide meaningful results. Despite this weak point, we believe that our results could provide helpful information to interventional cardiologists in terms of current real-world information showing long-term effects of statin treatment according to the different renal function groups. 

This study had other limitations. First, there may have been some underreporting and/or missing data and selection bias because this was a nonrandomized study. Second, although microalbuminuria is an early marker of chronic renal damage and a risk factor of cardiovascular disease [44], there was likely some misclassification of study groups due to the lack of information concerning the total amount of proteinuria and the presence or absence of microalbuminuria. Third, the estimation of renal function was based on a single measurement of eGFR at the time of presentation to the hospital. However, there is a possibility that eGFR may have worsened during the follow-up period. Unfortunately, we could not provide follow-up eGFR values because of a limitation of these registry data. Fourth, according to the current guidelines [7], the treatment goal is an LDL cholesterol level <1.8 mmol/L (<70 mg/dL) or at least 50% reduction in LDL cholesterol if the baseline LDL cholesterol level is 1.8–3.5 mmol/L. However, information regarding the follow-up levels of blood LDL cholesterol was incomplete in our registry data. This is a major shortcoming of this study and may be an important bias. Fifth, because the registry data did not include detailed or complete data on prescription doses, long-term adherence, discontinuation and drug-related adverse events, we could not provide this information during the follow-up period, which could have caused bias. Sixth, despite multivariable analyses, the variables that were not included in the data registry might have affected the study outcome. Seventh, because statins have a longer duration of use, the 2-year follow-up period in this study was relatively short for estimating long-term clinical outcomes. Eighth, because this retrospective study enrolled patients who underwent PCI between January 2006 and June 2015, this broad timeframe could have affected the clinical outcomes. Finally, during a 2-year follow-period, patients experienced definite or probable stent thrombosis (ST). Both in group A (group A1 vs. A2 vs. A3 vs. A4 = 37/6847 (0.5%) vs. 46/6557 (0.7%) vs. 22/2144 (1.0%) vs. 5/507 (1.0%), *p* = 0.091) and B (9/889 (1.0%) vs. 7/1227 (0.6%) vs. 4/537 (0.7%) vs. 4/537 (0.7%) vs. 1/167 (0.6%), *p* = 0.702, respectively), the cumulative incidences of ST were very low. Therefore, although ST is an important major determinant variable in patients with AMI [18], we inevitably could not include this variable as an endpoint in our study. 

## 5. Conclusions

In the era of newer-generation DESs, although statin treatment was effective in reducing mortality, this beneficial effect was diminished in accordance with the deterioration of baseline renal function in patients with AMI who underwent successful PCI. These results could be helpful in understanding the current real-world effects of statins on patients with AMI with different renal functions. However, more large-scale, long-term follow-up studies are warranted to confirm these results.

## Figures and Tables

**Figure 1 jcm-10-03504-f001:**
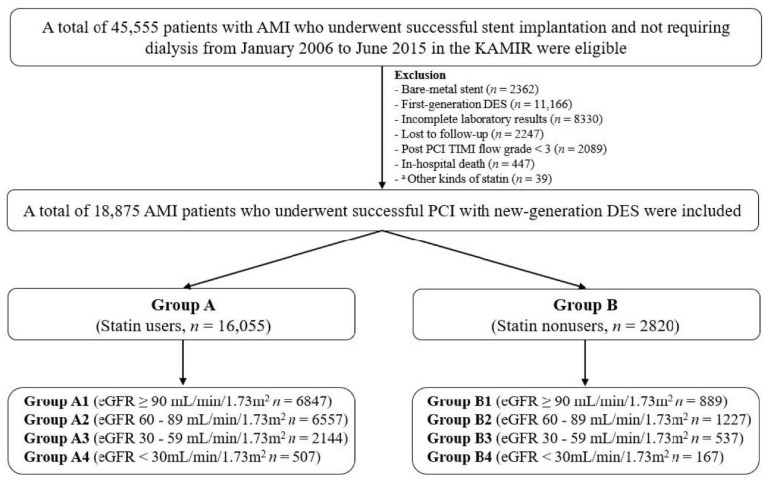
Flowchart. AMI, acute myocardial infarction; KAMIR, Korea AMI Registry; PCI, percutaneous coronary intervention; DES, drug-eluting stent; eGFR, estimated glomerular filtration rate. ^a^ Statins except for atorvastatin, rosuvastatin, simvastatin, pitavastatin, pravastatin and fluvastatin.

**Figure 2 jcm-10-03504-f002:**
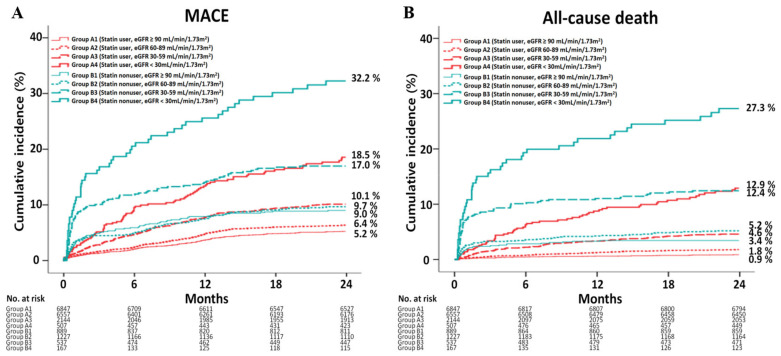
Kaplan–Meier analysis for the MACE (**A**), all-cause death (**B**), cardiac death (**C**), recurrent MI (**D**) and any repeat revascularization (**E**) during a 2-year follow-up period.

**Table 1 jcm-10-03504-t001:** Baseline characteristics of statin users.

Variables	Total (*n* = 16,055)	Group A1 eGFR ≥ 90 mL/min/1.73 m^2^ (*n* = 6847)	Group A2 eGFR 60–89 mL/min/1.73 m^2^ (*n* = 6557)	Group A3 eGFR 30–59 mL/min/1.73 m^2^ (*n* = 2144)	Group A4 eGFR < 30 mL/min/1.73 m^2^ (*n* = 507)	*p* Value
Male, *n* (%)	12,053 (75.1)	5536 (80.9)	4988 (76.1)	1225 (57.1)	304 (60.0)	<0.001
Age (years)	63.0 ± 12.3	58.9 ± 11.5	64.3 ± 12.1	70.8 ± 10.4	68.9 ± 10.9	<0.001
LVEF (%)	52.8 ± 10.8	54.0 ± 10.0	52.8 ± 10.7	50.0 ± 12.3	48.1 ± 11.9	<0.001
BMI (kg/m^2^)	24.2 ± 3.2	24.3 ± 3.2	24.2 ± 3.2	23.9 ± 3.2	23.5 ±3.4	<0.001
SBP (mmHg)	131.6 ± 27.2	133.4 ± 25.0	131.4 ± 27.7	126.6 ± 30.2	132.2 ± 33.3	<0.001
DBP (mmHg)	79.9 ± 16.2	81.7 ± 15.3	79.6 ± 16.4	75.7 ± 17.1	77.4 ± 18.0	<0.001
Cardiogenic shock, *n* (%)	575 (3.6)	124 (1.8)	247 (3.8)	160 (7.5)	44 (8.7)	<0.001
CPR on admission, *n* (%)	557 (3.5)	186 (2.7)	218 (3.3)	129 (6.0)	24 (4.7)	<0.001
Killip class III/IV, *n* (%)	1454 (9.1)	322 (4.7)	578 (8.8)	416 (19.4)	138 (27.2)	<0.001
STEMI, *n* (%)	8737 (54.4)	3640 (53.2)	3764 (57.4)	1160 (54.1)	173 (34.1)	<0.001
Primary PCI, *n* (%)	8424 (96.4)	3511 (96.5)	3633 (96.5)	1115 (96.1)	165 (95.4)	0.809
NSTEMI, *n* (%)	7318 (45.6)	3207 (46.8)	2793 (42.6)	984 (45.9)	334 (65.9)	<0.001
PCI within 24 h, *n* (%)	6303 (86.1)	2865 (89.3)	2384 (85.4)	791 (80.4)	263 (78.7)	<0.001
Hypertension, *n* (%)	7761 (48.3)	2622 (38.3)	3238 (49.4)	1489 (69.4)	412 (81.3)	<0.001
Diabetes mellitus, *n* (%)	4201 (26.2)	1446 (21.1)	1534 (23.4)	903 (42.1)	318 (62.7)	<0.001
Dyslipidemia, *n* (%)	1936 (12.1)	792 (11.6)	815 (12.4)	271 (12.6)	58 (11.4)	0.351
Previous MI, *n* (%)	661 (4.1)	214 (3.1)	261 (4.0)	140 (6.5)	46 (9.1)	<0.001
Previous PCI, *n* (%)	1008 (6.3)	327 (4.8)	394 (6.0)	215 (10.0)	72 (14.2)	<0.001
Previous CABG, *n* (%)	72 (0.4)	18 (0.3)	23 (0.4)	23 (1.1)	8 (1.6)	<0.001
Previous HF, *n* (%)	150 (0.9)	24 (0.4)	52 (0.8)	54 (2.5)	20 (3.9)	<0.001
Previous CVA, *n* (%)	947 (5.9)	252 (3.7)	390 (5.9)	241 (11.2)	64 (12.6)	<0.001
Current smokers, *n* (%)	6957 (43.3)	3560 (52.0)	2721 (41.5)	566 (26.4)	110 (21.7)	<0.001
Peak CK-MB (mg/dL)	121.2 ± 186.0	123.9 ± 178.0	128.1 ± 206.3	103.5 ± 151.0	70.8 ± 127.9	<0.001
Peak troponin-I (ng/mL)	47.0 ± 128.3	41.9 ± 69.5	48.6 ± 138.4	46.4 ± 94.5	97.1 ± 204.6	<0.001
NT-ProBNP (pg/mL)	1935.2 ± 4876.9	1258.8 ± 1542.6	1543.0 ± 2163.7	3188.7 ± 5091.2	9248.5 ± 8231.4	<0.001
High-sensitivity CRP (mg/dL)	7.5 ± 37.5	6.1 ± 31.8	8.3 ± 42.5	8.9 ± 38.6	10.1 ± 32.4	<0.001
Serum creatinine (mg/L)	1.1 ± 1.1	0.7 ± 0.1	1.0 ± 0.2	1.4 ± 0.3	5.1 ± 4.6	<0.001
eGFR (mL/min/1.73 m^2^)	87.5 ± 35.5	115.3 ± 33.7	76.3 ± 8.5	49.2 ± 8.0	16.5 ± 8.5	<0.001
Blood glucose (mg/L)	165.8 ± 76.7	155.2 ± 63.4	163.2 ± 70.4	197.3 ± 101.1	209.0 ± 129.7	<0.001
Total cholesterol (mg/dL)	184.3 ± 45.4	186.4 ± 43.1	186.0 ± 45.6	177.1 ± 47.9	162.4 ± 52.3	<0.001
Triglyceride (mg/L)	137.0 ± 114.0	140.0 ± 118.0	137.6 ± 114.9	128.3 ± 96.9	125.7 ± 111.3	<0.001
HDL cholesterol (mg/L)	43.3 ± 14.7	43.3 ± 13.3	43.8 ± 15.5	42.7 ± 16.5	39.3 ± 12.7	<0.001
LDL cholesterol (mg/L)	116.4 ± 40.7	119.4 ± 41.2	116.9 ± 37.5	109.5 ± 46.5	95.9 ± 38.8	<0.001
Discharge medications, *n* (%)						
Aspirin, *n* (%)	15,954 (99.4)	6814 (99.5)	6515 (99.4)	2127 (99.2)	498 (98.2)	0.003
Clopidogrel, *n* (%)	13,600 (84.7)	5528 (80.7)	5674 (86.5)	1941 (90.5)	457 (90.1)	<0.001
Ticagrelor, *n* (%)	1565 (9.7)	786 (11.5)	599 (9.1)	145 (6.8)	35 (6.9)	<0.001
Prasugrel, *n* (%)	890 (5.5)	533 (7.8)	284 (4.3)	58 (2.7)	15 (3.0)	<0.001
Cilostazole, *n* (%)	2903 (18.1)	1146 (16.7)	1226 (18.7)	434 (20.2)	97 (19.1)	0.001
ACEIs, *n* (%)	8977 (55.9)	3784 (55.3)	3893 (59.4)	1122 (52.3)	178 (35.1)	<0.001
ARBs, *n* (%)	4580 (28.5)	2023 (29.5)	1665 (25.4)	677 (31.6)	215 (42.4)	<0.001
BBs, *n* (%)	13,856 (86.3)	5984 (87.4)	5666 (86.4)	1775 (82.8)	431 (85.0)	<0.001
CCBs, *n* (%)	960 (6.0)	313 (4.6)	389 (5.9)	177 (8.3)	81 (16.0)	<0.001
Statin, *n* (%)						
Atorvastatin, *n* (%)	8636 (53.8)	3556 (51.9)	3552 (54.2)	1202 (56.1)	326 (64.3)	<0.001
Rosuvastatin, *n* (%)	5003 (31.2)	2254 (32.9)	2025 (30.9)	608 (28.4)	116 (22.9)	<0.001
Simvastatin, *n* (%)	949 (5.9)	378 (5.5)	412 (6.3)	133 (6.2)	26 (5.1)	0.222
Pitavastatin, *n* (%)	1211 (7.5)	559 (8.1)	478 (7.3)	154 (7.2)	20 (3.9)	0.003
Pravastatin, *n* (%)	214 (1.3)	86 (1.3)	76 (1.2)	40 (1.9)	12 (2.4)	0.014
Fluvastatin, *n* (%)	42 (0.3)	14 (0.2)	14 (0.2)	7 (0.3)	7 (1.4)	<0.001
Infarct-related artery						
Left main, *n* (%)	294 (1.8)	111 (1.6)	106 (1.6)	51 (2.4)	26 (5.1)	<0.001
LAD, *n* (%)	7704 (48.0)	3440 (50.2)	3126 (47.7)	917 (42.8)	221 (43.6)	<0.001
LCx, *n* (%)	2713 (16.9)	1232 (18.0)	1085 (16.5)	324 (15.1)	72 (14.2)	0.003
RCA, *n* (%)	5344 (33.3)	2064 (30.1)	2240 (34.2)	852 (39.7)	188 (37.1)	<0.001
Treated vessel						
Left main, *n* (%)	468 (2.9)	175 (2.6)	179 (2.7)	82 (3.8)	32 (6.3)	<0.001
LAD, *n* (%)	9348 (58.2)	4105 (60.0)	3770 (57.5)	1175 (54.8)	298 (58.8)	<0.001
LCx, *n* (%)	4239 (26.4)	1875 (27.4)	1675 (25.5)	568 (26.5)	121 (23.9)	0.056
RCA, *n* (%)	6401 (39.9)	2509 (36.6)	2672 (40.8)	997 (46.5)	223 (44.0)	<0.001
ACC/AHA lesion type						
Type B1, *n* (%)	2180 (13.6)	827 (12.1)	978 (14.9)	303 (14.1)	72 (14.2)	<0.001
Type B2, *n* (%)	5677 (35.4)	2643 (38.6)	2140 (32.6)	720 (33.6)	174 (34.3)	<0.001
Type C, *n* (%)	7249 (45.2)	3026 (44.2)	3002 (45.8)	983 (45.8)	238 (46.9)	0.198
Extent of CAD						
Single-vessel, *n* (%)	8150 (50.8)	3827 (55.9)	3250 (49.6)	875 (40.8)	198 (39.1)	<0.001
2-vessel, *n* (%)	4712 (29.3)	1925 (28.1)	1968 (30.0)	671 (31.3)	148 (29.2)	0.016
≥3-vessel, *n* (%)	3193 (19.9)	1095 (16.0)	1339 (20.4)	598 (27.9)	161 (31.8)	<0.001
Pre-PCI TIMI 0/1, *n* (%)	9308 (58.0)	3857 (56.3)	3966 (60.5)	1258 (58.7)	227 (44.8)	<0.001
Type of stent						
ZES, *n* (%)	5591 (34.8)	2219 (32.4)	2404 (36.7)	801 (37.4)	167 (32.9)	<0.001
EES, *n* (%)	7888 (49.1)	3335 (48.7)	3187 (48.6)	1092 (50.9)	274 (54.0)	0.031
BES, *n* (%)	2418 (15.1)	1161 (17.0)	944 (14.4)	249 (11.6)	64 (12.6)	<0.001
Others, *n* (%)	475 (3.0)	264 (3.9)	160 (2.4)	43 (2.0)	8 (1.6)	<0.001
IVUS, *n* (%)	3619 (19.7)	1358 (19.8)	1321 (20.1)	401 (18.7)	89 (17.6)	0.295
OCT, *n* (%)	113 (0.7)	61 (0.9)	47 (0.7)	4 (0.2)	1 (0.2)	0.004
FFR, *n* (%)	211 (1.3)	135 (2.0)	69 (1.0)	5 (0.2)	2 (0.4)	<0.001
Stent diameter (mm)	3.15 ± 0.42	3.16 ± 0.42	3.16 ± 0.43	3.11 ± 0.42	3.09 ± 0.43	<0.001
Stent length (mm)	27.1 ± 11.7	26.9 ± 11.6	26.9 ± 11.5	27.8 ± 12.3	29.0 ± 13.7	<0.001
Number of stents	1.48 ± 0.79	1.45 ± 0.77	1.48 ± 0.79	1.54 ± 0.82	1.60 ± 0.87	<0.001

Values are means ± SDs or numbers and percentages. The *p* values for continuous data were obtained from the analysis of variance. The *p* values for categorical data were obtained from the chi-square or Fisher’s exact test. eGFR, estimated glomerular filtration rate; LVEF, left ventricular ejection fraction; BMI, body mass index; SBP, systolic blood pressure; DBP, diastolic blood pressure; CPR, cardiopulmonary resuscitation; STEMI, ST-segment elevation myocardial infarction; PCI, percutaneous coronary intervention; NSTEMI, non-STEMI; MI, myocardial infarction; CABG, coronary artery bypass graft; HF, heart failure; CVA, cerebrovascular accidents; CK-MB, creatine kinase myocardial band; NT-ProBNP, N-terminal pro-brain natriuretic peptide; CRP, C-reactive protein; HDL, high-density lipoprotein; LDL, low-density lipoprotein; ACEIs, angiotensin-converting enzyme inhibitors; ARBs, angiotensin receptor blockers; BBs, beta blockers; CCBs, calcium channel blockers; LAD, left anterior descending coronary artery; LCx, left circumflex coronary artery; RCA, right coronary artery; ACC/AHA, American College of Cardiology/American Heart Association; CAD, coronary artery disease; TIMI, thrombolysis in myocardial infarction; ZES, zotarolimus-eluting stent; EES, everolimus-eluting stent; BES, biolimus-eluting stent; IVUS, intravascular ultrasound; OCT, optical coherence tomography; FFR, fractional flow reserve.

**Table 2 jcm-10-03504-t002:** Baseline characteristics in statin nonusers.

Variables	Total (*n* = 2820)	Group B1 eGFR ≥ 90 mL/min/1.73 m^2^ (*n* = 889)	Group B2 eGFR 60–89 mL/min/1.73 m^2^ (*n* = 1227)	Group B3 eGFR 30–59 mL/min/1.73 m^2^ (*n* = 537)	Group B4 eGFR < 30 mL/min/1.73 m^2^ (*n* = 167)	*p* Value
Male, *n* (%)	2067 (73.3)	730 (82.1)	921 (75.1)	317 (59.0)	99 (59.3)	<0.001
Age (years)	64.1 ± 12.4	58.8 ± 11.6	64.3 ± 12.2	70.8 ± 10.5	69.8 ± 10.4	<0.001
LVEF (%)	50.9 ± 12.2	52.9 ± 11.5	51.5 ± 11.7	48.4 ± 13.1	44.2 ± 12.8	<0.001
BMI (kg/m^2^)	23.8 ± 3.3	23.9 ± 3.4	23.8 ± 3.2	23.7 ± 3.5	23.1 ±3.4	0.055
SBP (mmHg)	128.6 ± 29.1	132.2 ± 26.7	129.7 ± 27.6	120.0 ± 32.1	128.0 ± 25.8	<0.001
DBP (mmHg)	77.9 ± 16.8	80.7 ± 15.3	78.9 ± 16.4	72.0 ± 17.5	73.7 ± 19.6	<0.001
Cardiogenic shock, *n* (%)	153 (5.4)	22 (2.5)	48 (3.9)	58 (10.8)	25 (15.0)	<0.001
CPR on admission, *n* (%)	151 (5.4)	30 (3.4)	59 (4.8)	51 (9.5)	11 (6.6)	<0.001
Killip class III/IV, *n* (%)	393 (13.9)	53 (6.0)	140 (11.4)	146 (27.2)	54 (32.3)	<0.001
STEMI, *n* (%)	1630 (57.8)	512 (57.6)	744 (60.6)	309 (57.5)	65 (38.9)	<0.001
Primary PCI, *n* (%)	1551 (95.2)	485 (94.7)	705 (94.8)	297 (96.1)	64 (98.5)	0.454
NSTEMI, *n* (%)	1190 (42.2)	377 (42.4)	483(39.4)	228 (42.5)	102 (61.1)	<0.001
PCI within 24 h	912 (76.6)	299 (79.3)	366 (75.8)	170 (74.6)	77 (75.5)	0.507
Hypertension, *n* (%)	1391 (49.3)	318 (35.8)	598 (48.7)	340 (63.3)	135 (80.8)	<0.001
Diabetes mellitus, *n* (%)	867 (30.7)	233 (26.2)	309 (25.2)	216 (40.2)	109 (65.3)	<0.001
Dyslipidemia, *n* (%)	249 (8.8)	75 (8.4)	108 (8.8)	48 (8.9)	18 (10.8)	0.809
Previous MI, *n* (%)	96 (3.4)	21 (2.4)	33 (2.7)	27 (5.0)	15 (9.0)	<0.001
Previous PCI, *n* (%)	186 (6.6)	44 (4.9)	73 (8.9)	52 (9.7)	17 (10.2)	0.001
Previous CABG, *n* (%)	12 (0.4)	2 (0.2)	3 (0.2)	6 (1.0)	1 (0.6)	0.047
Previous HF, *n* (%)	58 (2.1)	9 (1.0)	19 (1.5)	19 (3.5)	11 (6.6)	<0.001
Previous CVA, *n* (%)	193 (6.8)	36 (4.0)	75 (6.1)	57 (10.6)	25 (15.0)	<0.001
Current smokers, *n* (%)	1157 (41.0)	471 (53.0)	498 (40.6)	158 (29.4)	30 (18.0)	<0.001
Peak CK-MB (mg/dL)	144.5 ± 319.5	127.7 ± 154.2	168.6 ± 442.3	133.8 ± 201.7	91.7 ± 144.5	0.002
Peak troponin-I (ng/mL)	46.8 ± 85.9	42.4 ± 61.2	48.2 ± 87.5	48.2 ± 92.3	55.3 ± 145.3	0.215
NT-ProBNP (pg/mL)	2487.6 ± 4085.6	1499.8 ± 2460.2	1599.3 ± 2089.5	3802.2 ± 5998.2	9948.8 ± 9432.6	<0.001
High-sensitivity CRP (mg/dL)	9.8 ± 38.1	9.1 ± 34.5	8.3 ± 33.9	12.7 ± 47.9	15.2 ± 48.2	<0.001
Serum creatinine (mg/L)	1.2 ± 1.4	0.7 ± 0.1	1.0 ± 1.2	1.4 ± 0.3	5.1 ± 3.7	<0.001
eGFR (mL/min/1.73 m^2^)	76.8 ± 38.3	114.7 ± 44.3	75.6 ± 8.5	48.6 ± 8.2	15.8 ± 8.2	<0.001
Blood glucose (mg/L)	177.7 ± 85.5	163.4 ± 65.7	169.6 ± 75.1	207.9 ± 108.3	217.0 ± 124.4	<0.001
Total cholesterol (mg/dL)	174.4 ± 43.5	180.4 ± 44.1	178.1 ± 41.5	163.9 ± 41.6	148.2 ± 45.6	<0.001
Triglyceride (mg/L)	132.4 ± 114.0	142.0 ± 122.3	135.5 ± 125.1	113.2 ± 73.0	117.5 ± 68.7	<0.001
HDL cholesterol (mg/L)	43.0 ± 15.6	43.1 ± 11.7	43.8 ± 17.1	42.5 ± 17.6	37.9 ± 14.8	<0.001
LDL cholesterol (mg/L)	108.2 ± 39.2	112.5 ± 37.6	111.3 ± 38.1	100.7 ± 41.8	83.9 ± 36.4	<0.001
Discharge medications, *n* (%)						
Aspirin, *n* (%)	2714 (96.2)	848 (95.5)	1192 (97.1)	518 (96.5)	156 (93.4)	0.040
Clopidogrel, *n* (%)	2726 (96.6)	843 (94.8)	1200 (97.8)	523 (91.1)	160 (95.8)	<0.001
Ticagrelor, *n* (%)	53 (1.9)	22 (2.5)	16 (1.3)	10 (1.9)	5 (3.0)	0.169
Prasugrel, *n* (%)	41 (1.5)	24 (2.7)	11 (0.9)	4 (0.7)	2 (1.2)	0.003
Cilostazole, *n* (%)	483 (17.1)	139 (15.6)	241 (19.6)	82 (15.3)	21 (12.6)	0.014
ACEIs, *n* (%)	1127 (40.0)	356 (40.0)	539 (43.9)	189 (35.2)	43 (25.7)	<0.001
ARBs, *n* (%)	677 (24.0)	238 (26.8)	280 (22.8)	124 (23.1)	35 (21.0)	0.123
BBs, *n* (%)	1838 (65.2)	605 (68.1)	839 (68.4)	315 (58.7)	79 (47.3)	<0.001
CCBs, *n* (%)	176 (6.2)	44 (4.9)	70 (5.7)	41 (7.6)	21 (12.6)	0.001
Infarct-related artery						
Left main, *n* (%)	58 (2.1)	20 (2.2)	14 (1.1)	16 (3.0)	8 (4.8)	0.003
LAD, *n* (%)	1324 (47.0)	434 (48.8)	593 (48.3)	228 (42.4)	69 (41.3)	0.037
LCx, *n* (%)	451 (16.0)	163 (18.3)	190 (15.5)	69 (12.8)	29 (17.4)	0.045
RCA, *n* (%)	987 (35.0)	272 (30.6)	430 (35.0)	224 (41.7)	61 (36.5)	<0.001
Treated vessel						
Left main, *n* (%)	77 (2.7)	26 (2.9)	21 (1.7)	17 (3.2)	13 (7.8)	<0.001
LAD, *n* (%)	1552 (55.0)	508 (57.1)	676 (55.1)	278 (51.8)	90 (53.9)	0.261
LCx, *n* (%)	670 (23.8)	228 (25.6)	274 (22.3)	127 (23.6)	41 (24.6)	0.363
RCA, *n* (%)	1132 (40.1)	318 (35.8)	492 (40.1)	253 (47.1)	99 (41.3)	<0.001
ACC/AHA lesion type						
Type B1, *n* (%)	509 (18.0)	161 (18.1)	223 (18.2)	109 (20.3)	16 (9.6)	0.019
Type B2, *n* (%)	857 (30.4)	296 (33.3)	371 (30.2)	133 (24.8)	57 (34.1)	0.005
Type C, *n* (%)	1141 (40.5)	340 (38.2)	494 (40.3)	224 (41.7)	83 (49.7)	0.044
Extent of CAD						
Single-vessel, *n* (%)	1369 (48.5)	481 (54.1)	614 (50.0)	211 (39.3)	63 (37.7)	<0.001
2-vessel, *n* (%)	818 (29.0)	242 (27.2)	352 (28.7)	172 (32.0)	52 (31.1)	0.242
≥3-vessel, *n* (%)	633 (22.4)	166 (18.7)	261 (21.3)	154 (28.7)	52 (31.1)	<0.001
Pre-PCI TIMI 0/1, *n* (%)	1745 (61.9)	544 (61.2)	788 (64.2)	330 (61.5)	83 (49.7)	0.004
Type of stent						
ZES, *n* (%)	1171 (41.5)	347 (39.0)	526 (42.9)	233 (43.4)	65 (38.9)	0.219
EES, *n* (%)	1269 (45.0)	389 (43.8)	556 (45.3)	245 (45.9)	79 (47.3)	0.790
BES, *n* (%)	315 (11.2)	123 (13.8)	122 (9.9)	52 (9.7)	18 (10.8)	0.024
Others, *n* (%)	84 (3.0)	34 (3.8)	31 (2.5)	12 (2.2)	7 (4.2)	0.176
IVUS, *n* (%)	502 (17.8)	169 (19.0)	218 (17.8)	85 (15.8)	30 (18.0)	0.509
OCT, *n* (%)	17 (0.6)	7 (0.8)	7 (0.6)	3 (0.6)	0	0.669
FFR, *n* (%)	7 (0.2)	3 (0.3)	0	3 (0.6)	1 (0.6)	0.100
Stent diameter (mm)	3.16 ± 0.44	3.18 ± 0.43	3.17 ± 0.43	3.11 ± 0.44	3.11 ± 0.44	0.019
Stent length (mm)	25.0 ± 9.4	24.1 ± 8.1	24.7 ± 8.5	26.0 ± 11.1	28.1 ± 14.4	<0.001
Number of stents	1.43 ± 0.76	1.40 ± 0.72	1.40 ± 0.75	1.53 ± 0.82	1.51 ± 0.76	0.004

Values are means ± SDs or numbers and percentages. The *p* values for continuous data were obtained from the analysis of variance. The *p* values for categorical data were obtained from the chi-square or Fisher’s exact test. eGFR, estimated glomerular filtration rate; LVEF, left ventricular ejection fraction; BMI, body mass index; SBP, systolic blood pressure; DBP, diastolic blood pressure; CPR, cardiopulmonary resuscitation; STEMI, ST-segment elevation myocardial infarction; PCI, percutaneous coronary intervention; NSTEMI, non-STEMI; MI, myocardial infarction; CABG, coronary artery bypass graft; HF, heart failure; CVA, cerebrovascular accidents; CK-MB, creatine kinase myocardial band; NT-ProBNP, N-terminal pro-brain natriuretic peptide; CRP, C-reactive protein; HDL, high-density lipoprotein; LDL, low-density lipoprotein; ACEIs, angiotensin-converting enzyme inhibitors; ARBs, angiotensin receptor blockers; BBs, beta blockers; CCBs, calcium channel blockers; LAD, left anterior descending coronary artery; LCx, left circumflex coronary artery; RCA, right coronary artery; ACC/AHA, American College of Cardiology/American Heart Association; CAD, coronary artery disease; TIMI, thrombolysis in myocardial infarction; ZES, zotarolimus-eluting stent; EES, everolimus-eluting stent; BES, biolimus-eluting stent; IVUS, intravascular ultrasound; OCT, optical coherence tomography; FFR, fractional flow reserve.

**Table 3 jcm-10-03504-t003:** Baseline characteristics between statin users and nonusers 1.

Variables	Group A1 eGFR ≥ 90 mL/min/1.73 m^2^ (*n* = 6847)	Group B1 eGFR ≥ 90 mL/min/1.73 m^2^ (*n* = 889)	*p* Value	Group A2 eGFR 60–89 mL/min/1.73 m^2^ (*n* = 6557)	Group B2 eGFR 60–89 mL/min/1.73 m^2^ (*n* = 1227)	*p* Value
Male, *n* (%)	5536 (80.9)	730 (82.1)	0.367	4988 (76.1)	921 (75.1)	0.448
Age (years)	58.9 ± 11.5	58.8 ± 11.6	0.848	64.3 ± 12.1	64.3 ± 12.2	0.915
LVEF (%)	54.0 ± 10.0	52.9 ± 11.5	0.007	52.8 ± 10.7	51.5 ± 11.7	0.001
BMI (kg/m^2^)	24.2 ± 3.2	23.9 ± 3.4	0.009	24.2 ± 3.2	23.8 ± 3.2	<0.001
SBP (mmHg)	133.4 ± 25.0	132.2 ± 26.7	0.221	131.4 ± 27.7	129.7 ± 27.6	0.051
DBP (mmHg)	81.7 ± 15.3	80.7 ± 15.3	0.074	79.6 ± 16.4	78.9 ± 16.4	0.220
Cardiogenic shock, *n* (%)	124 (1.8)	22 (2.5)	0.171	247 (3.8)	48 (3.9)	0.807
CPR on admission, *n* (%)	186 (2.7)	30 (3.4)	0.263	218 (3.3)	59 (4.8)	0.010
Killip class III/IV, *n* (%)	322 (4.7)	53 (6.0)	0.100	578 (8.8)	140 (11.4)	0.004
STEMI, *n* (%)	3640 (53.2)	512 (57.6)	0.013	3764 (57.4)	744 (60.6)	0.035
Primary PCI, *n* (%)	3511 (96.5)	485 (94.7)	0.054	3633 (96.5)	705 (94.8)	0.021
NSTEMI, *n* (%)	3207 (46.8)	377 (42.4)	0.013	2793 (42.6)	483(39.4)	0.035
PCI within 24 h	2865 (89.3)	299 (79.3)	<0.001	2384 (85.4)	366 (75.8)	<0.001
Hypertension, *n* (%)	2622 (38.3)	318 (35.8)	0.145	3238 (49.4)	598 (48.7)	0.678
Diabetes mellitus, *n* (%)	1446 (21.1)	233 (26.2)	0.001	1534 (23.4)	309 (25.2)	0.176
Dyslipidemia, *n* (%)	792 (11.6)	75 (8.4)	0.005	815 (12.4)	108 (8.8)	<0.001
Previous MI, *n* (%)	214 (3.1)	21 (2.4)	0.212	261 (4.0)	33 (2.7)	0.029
Previous PCI, *n* (%)	327 (4.8)	44 (4.9)	0.820	394 (6.0)	73 (8.9)	0.936
Previous CABG, *n* (%)	18 (0.3)	2 (0.2)	0.834	23 (0.4)	3 (0.2)	0.788
Previous HF, *n* (%)	24 (0.4)	9 (1.0)	0.004	52 (0.8)	19 (1.5)	0.011
Previous CVA, *n* (%)	252 (3.7)	36 (4.0)	0.585	390 (5.9)	75 (6.1)	0.823
Current smokers, *n* (%)	3560 (52.0)	471 (53.0)	0.579	2721 (41.5)	498 (40.6)	0.552
Peak CK-MB (mg/dL)	123.9 ± 178.0	127.7 ± 154.2	0.538	128.1 ± 206.3	168.6 ± 442.3	0.002
Peak troponin-I (ng/mL)	41.9 ± 69.5	42.4 ± 61.2	0.823	48.6 ± 138.4	48.2 ± 87.5	0.905
NT-ProBNP (pg/mL)	1258.8 ± 1542.6	1499.8 ± 2460.2	0.004	1543.0 ± 2163.7	1599.3 ± 2089.5	0.389
High-sensitivity CRP (mg/dL)	6.1 ± 31.8	9.1 ± 34.5	0.012	8.3 ± 42.5	8.3 ± 33.9	0.972
Serum creatinine (mg/L)	0.7 ± 0.1	0.7 ± 0.1	0.062	1.0 ± 0.2	1.0 ± 1.2	0.245
eGFR (mL/min/1.73 m^2^)	115.3 ± 33.7	114.7 ± 44.3	0.725	76.3 ± 8.5	75.6 ± 8.5	0.060
Blood glucose (mg/L)	155.2 ± 63.4	163.4 ± 65.7	0.001	163.2 ± 70.4	169.6 ± 75.1	0.007
Total cholesterol (mg/dL)	186.4 ± 43.1	180.4 ± 44.1	<0.001	186.0 ± 45.6	178.1 ± 41.5	<0.001
Triglyceride (mg/L)	140.0 ± 118.0	142.0 ± 122.3	0.639	137.6 ± 114.9	135.5 ± 125.1	0.592
HDL cholesterol (mg/L)	43.3 ± 13.3	43.1 ± 11.7	0.676	43.8 ± 15.5	43.8 ± 17.1	0.943
LDL cholesterol (mg/L)	119.4 ± 41.2	112.5 ± 37.6	<0.001	116.9 ± 37.5	111.3 ± 38.1	<0.001
Discharge medications, *n* (%)						
Aspirin, *n* (%)	6814 (99.5)	848 (95.5)	<0.001	6515 (99.4)	1192 (97.1)	<0.001
Clopidogrel, *n* (%)	5528 (80.7)	843 (94.8)	<0.001	5674 (86.5)	1200 (97.8)	<0.001
Ticagrelor, *n* (%)	786 (11.5)	22 (2.5)	<0.001	599 (9.1)	16 (1.3)	<0.001
Prasugrel, *n* (%)	533 (7.8)	24 (2.7)	<0.001	284 (4.3)	11 (0.9)	<0.001
Cilostazole, *n* (%)	1146 (16.7)	139 (15.6)	0.406	1226 (18.7)	241 (19.6)	0.438
ACEIs, *n* (%)	3784 (55.3)	356 (40.0)	<0.001	3893 (59.4)	539 (43.9)	<0.001
ARBs, *n* (%)	2023 (29.5)	238 (26.8)	0.087	1665 (25.4)	280 (22.8)	0.057
BBs, *n* (%)	5984 (87.4)	605 (68.1)	<0.001	5666 (86.4)	839 (68.4)	<0.001
CCBs, *n* (%)	313 (4.6)	44 (4.9)	0.613	389 (5.9)	70 (5.7)	0.792
Statin, *n* (%)						
Atorvastatin, *n* (%)	3556 (51.9)			3552 (54.2)		
Rosuvastatin, *n* (%)	2254 (32.9)			2025 (30.9)		
Simvastatin, *n* (%)	378 (5.5)			412 (6.3)		
Pitavastatin, *n* (%)	559 (8.1)			478 (7.3)		
Pravastatin, *n* (%)	86 (1.3)			76 (1.2)		
Fluvastatin, *n* (%)	14 (0.2)			14 (0.2)		
Infarct-related artery						
Left main, *n* (%)	111 (1.6)	20 (2.2)	0.172	106 (1.6)	14 (1.1)	0.256
LAD, *n* (%)	3440 (50.2)	434 (48.8)	0.347	3126 (47.7)	593 (48.3)	0.747
LCx, *n* (%)	1232 (18.0)	163 (18.3)	0.803	1085 (16.5)	190 (15.5)	0.356
RCA, *n* (%)	2064 (30.1)	272 (30.6)	0.783	2240 (34.2)	430 (35.0)	0.550
Treated vessel						
Left main, *n* (%)	175 (2.6)	26 (2.9)	0.516	179 (2.7)	21 (1.7)	0.039
LAD, *n* (%)	4105 (60.0)	508 (57.1)	0.108	3770 (57.5)	676 (55.1)	0.119
LCx, *n* (%)	1875 (27.4)	228 (25.6)	0.273	1675 (25.5)	274 (22.3)	0.018
RCA, *n* (%)	2509 (36.6)	318 (35.8)	0.611	2672 (40.8)	492 (40.1)	0.669
ACC/AHA lesion type						
Type B1, *n* (%)	827 (12.1)	161 (18.1)	<0.001	978 (14.9)	223 (18.2)	0.004
Type B2, *n* (%)	2643 (38.6)	296 (33.3)	0.002	2140 (32.6)	371 (30.2)	0.103
Type C, *n* (%)	3026 (44.2)	340 (38.2)	0.001	3002 (45.8)	494 (40.3)	<0.001
Extent of CAD						
Single-vessel, *n* (%)	3827 (55.9)	481 (54.1)	0.313	3250 (49.6)	614 (50.0)	0.760
2-vessel, *n* (%)	1925 (28.1)	242 (27.2)	0.606	1968 (30.0)	352 (28.7)	0.351
≥3-vessel, *n* (%)	1095 (16.0)	166 (18.7)	0.042	1339 (20.4)	261 (21.3)	0.499
Pre-PCI TIMI 0/1, *n* (%)	3857 (56.3)	544 (61.2)	0.006	3966 (60.5)	788 (64.2)	0.014
Type of stent						
ZES, *n* (%)	2219 (32.4)	347 (39.0)	<0.001	2404 (36.7)	526 (42.9)	<0.001
EES, *n* (%)	3335 (48.7)	389 (43.8)	0.005	3187 (48.6)	556 (45.3)	0.034
BES, *n* (%)	1161 (17.0)	123 (13.8)	0.019	944 (14.4)	122 (9.9)	<0.001
Others, *n* (%)	264 (3.9)	34 (3.8)	0.964	160 (2.4)	31 (2.5)	0.841
IVUS, *n* (%)	1358 (19.8)	169 (19.0)	0.591	1321 (20.1)	218 (17.8)	0.056
OCT, *n* (%)	61 (0.9)	7 (0.8)	0.756	47 (0.7)	7 (0.6)	0.709
FFR, *n* (%)	135 (2.0)	3 (0.3)	0.001	69 (1.0)	0	<0.001
Stent diameter (mm)	3.16 ± 0.42	3.18 ± 0.43	0.344	3.16 ± 0.43	3.17 ± 0.43	0.274
Stent length (mm)	26.9 ± 11.6	24.1 ± 8.1	<0.001	26.9 ± 11.5	24.7 ± 8.5	<0.001
Number of stents	1.45 ± 0.77	1.40 ± 0.72	0.039	1.48 ± 0.79	1.40 ± 0.75	0.002

Values are means ± SDs or numbers and percentages. The *p* values for continuous data were obtained from the unpaired *t*-test. The *p* values for categorical data were obtained from the chi-square or Fisher’s exact test. eGFR, estimated glomerular filtration rate; LVEF, left ventricular ejection fraction; BMI, body mass index; SBP, systolic blood pressure; DBP, diastolic blood pressure; CPR, cardiopulmonary resuscitation; STEMI, ST-segment elevation myocardial infarction; PCI, percutaneous coronary intervention; NSTEMI, non-STEMI; MI, myocardial infarction; CABG, coronary artery bypass graft; HF, heart failure; CVA, cerebrovascular accidents; CK-MB, creatine kinase myocardial band; NT-ProBNP, N-terminal pro-brain natriuretic peptide; CRP, C-reactive protein; HDL, high-density lipoprotein; LDL, low-density lipoprotein; ACEIs, angiotensin-converting enzyme inhibitors; ARBs, angiotensin receptor blockers; BBs, beta blockers; CCBs, calcium channel blockers; LAD, left anterior descending coronary artery; LCx, left circumflex coronary artery; RCA, right coronary artery; ACC/AHA, American College of Cardiology/American Heart Association; CAD, coronary artery disease; TIMI, thrombolysis in myocardial infarction; ZES, zotarolimus-eluting stent; EES, everolimus-eluting stent; BES, biolimus-eluting stent; IVUS, intravascular ultrasound; OCT, optical coherence tomography; FFR, fractional flow reserve.

**Table 4 jcm-10-03504-t004:** Baseline characteristics between statin users and nonusers 2.

Variables	Group A3 eGFR 30–59 mL/min/1.73 m^2^ (*n* = 2144)	Group B3 eGFR 30–59 mL/min/1.73 m^2^ (*n* = 537)	*p* Value	Group A4 eGFR < 30 mL/min/1.73 m^2^ (*n* = 507)	Group B4 eGFR < 30 mL/min/1.73 m^2^ (*n* = 167)	*p* Value
Male, *n* (%)	1225 (57.1)	317 (59.0)	0.427	304 (60.0)	99 (59.3)	0.877
Age (years)	70.8 ± 10.4	70.8 ± 10.5	0.884	68.9 ± 10.9	69.8 ± 10.4	0.376
LVEF (%)	50.0 ± 12.3	48.4 ± 13.1	0.020	48.1 ± 11.9	44.2 ± 12.8	0.001
BMI (kg/m^2^)	23.9 ± 3.2	23.7 ± 3.5	0.414	23.5 ±3.4	23.1 ±3.4	0.276
SBP (mmHg)	126.6 ± 30.2	120.0 ± 32.1	<0.001	132.2 ± 33.3	128.0 ± 25.8	0.183
DBP (mmHg)	75.7 ± 17.1	72.0 ± 17.5	<0.001	77.4 ± 18.0	73.7 ± 19.6	0.035
Cardiogenic shock, *n* (%)	160 (7.5)	58 (10.8)	0.011	44 (8.7)	25 (15.0)	0.020
CPR on admission, *n* (%)	129 (6.0)	51 (9.5)	0.004	24 (4.7)	11 (6.6)	0.349
Killip class III/IV, *n* (%)	416 (19.4)	146 (27.2)	<0.001	138 (27.2)	54 (32.3)	0.204
STEMI, *n* (%)	1160 (54.1)	309 (57.5)	0.152	173 (34.1)	65 (38.9)	0.260
Primary PCI, *n* (%)	1115 (96.1)	297 (96.1)	0.997	165 (95.4)	64 (98.5)	0.266
NSTEMI, *n* (%)	984 (45.9)	228 (42.5)	0.152	334 (65.9)	102 (61.1)	0.260
PCI within 24 h	791 (80.4)	170 (74.6)	0.051	263 (78.7)	77 (75.5)	0.488
Hypertension, *n* (%)	1489 (69.4)	340 (63.3)	0.006	412 (81.3)	135 (80.8)	0.903
Diabetes mellitus, *n* (%)	903 (42.1)	216 (40.2)	0.434	318 (62.7)	109 (65.3)	0.553
Dyslipidemia, *n* (%)	271 (12.6)	48 (8.9)	0.017	58 (11.4)	18 (10.8)	0.815
Previous MI, *n* (%)	140 (6.5)	27 (5.0)	0.231	46 (9.1)	15 (9.0)	0.972
Previous PCI, *n* (%)	215 (10.0)	52 (9.7)	0.872	72 (14.2)	17 (10.2)	0.235
Previous CABG, *n* (%)	23 (1.1)	6 (1.0)	0.929	8 (1.6)	1 (0.6)	0.464
Previous HF, *n* (%)	54 (2.5)	19 (3.5)	0.194	20 (3.9)	11 (6.6)	0.157
Previous CVA, *n* (%)	241 (11.2)	57 (10.6)	0.759	64 (12.6)	25 (15.0)	0.437
Current smokers, *n* (%)	566 (26.4)	158 (29.4)	0.159	110 (21.7)	30 (18.0)	0.302
Peak CK-MB (mg/dL)	103.5 ± 151.0	133.8 ± 201.7	<0.001	70.8 ± 127.9	91.7 ± 144.5	0.076
Peak troponin-I (ng/mL)	46.4 ± 94.5	48.2 ± 92.3	0.700	97.1 ± 204.6	55.3 ± 145.3	0.445
NT-ProBNP (pg/mL)	3188.7 ± 5091.2	3802.2 ± 5998.2	0.016	9248.5 ± 8231.4	9948.8 ± 9432.6	0.589
High-sensitivity CRP (mg/dL)	8.9 ± 38.6	12.7 ± 47.9	0.093	10.1 ± 32.4	15.2 ± 48.2	0.209
Serum creatinine (mg/L)	1.4 ± 0.3	1.4 ± 0.3	0.074	5.1 ± 4.6	5.1 ± 3.7	0.965
eGFR (mL/min/1.73 m^2^)	49.2 ± 8.0	48.6 ± 8.2	0.130	16.5 ± 8.5	15.8 ± 8.2	0.305
Blood glucose (mg/L)	197.3 ± 101.1	207.9 ± 108.3	0.044	209.0 ± 129.7	217.0 ± 124.4	0.479
Total cholesterol (mg/dL)	177.1 ± 47.9	163.9 ± 41.6	<0.001	162.4 ± 52.3	148.2 ± 45.6	0.001
Triglyceride (mg/L)	128.3 ± 96.9	113.2 ± 73.0	<0.001	125.7 ± 111.3	117.5 ± 68.7	0.282
HDL cholesterol (mg/L)	42.7 ± 16.5	42.5 ± 17.6	0.828	39.3 ± 12.7	37.9 ± 14.8	0.302
LDL cholesterol (mg/L)	109.5 ± 46.5	100.7 ± 41.8	<0.001	95.9 ± 38.8	83.9 ± 36.4	0.001
Discharge medications, *n* (%)						
Aspirin, *n* (%)	2127 (99.2)	518 (96.5)	<0.001	498 (98.2)	156 (93.4)	0.001
Clopidogrel, *n* (%)	1941 (90.5)	523 (91.1)	0.798	457 (90.1)	160 (95.8)	0.606
Ticagrelor, *n* (%)	145 (6.8)	10 (1.9)	<0.001	35 (6.9)	5 (3.0)	0.087
Prasugrel, *n* (%)	58 (2.7)	4 (0.7)	0.006	15 (3.0)	2 (1.2)	0.266
Cilostazole, *n* (%)	434 (20.2)	82 (15.3)	0.008	97 (19.1)	21 (12.6)	0.060
ACEIs, *n* (%)	1122 (52.3)	189 (35.2)	<0.001	178 (35.1)	43 (25.7)	0.029
ARBs, *n* (%)	677 (31.6)	124 (23.1)	<0.001	215 (42.4)	35 (21.0)	<0.001
BBs, *n* (%)	1775 (82.8)	315 (58.7)	<0.001	431 (85.0)	79 (47.3)	<0.001
CCBs, *n* (%)	177 (8.3)	41 (7.6)	0.724	81 (16.0)	21 (12.6)	0.321
Statin, *n* (%)						
Atorvastatin, *n* (%)	1202 (56.1)			326 (64.3)		
Rosuvastatin, *n* (%)	608 (28.4)			116 (22.9)		
Simvastatin, *n* (%)	133 (6.2)			26 (5.1)		
Pitavastatin, *n* (%)	154 (7.2)			20 (3.9)		
Pravastatin, *n* (%)	40 (1.9)			12 (2.4)		
Fluvastatin, *n* (%)	7 (0.3)			7 (1.4)		
Infarct-related artery						
Left main, *n* (%)	51 (2.4)	16 (3.0)	0.425	26 (5.1)	8 (4.8)	0.863
LAD, *n* (%)	917 (42.8)	228 (42.4)	0.963	221 (43.6)	69 (41.3)	0.472
LCx, *n* (%)	324 (15.1)	69 (12.8)	0.195	72 (14.2)	29 (17.4)	0.320
RCA, *n* (%)	852 (39.7)	224 (41.7)	0.404	188 (37.1)	61 (36.5)	0.927
Treated vessel						
Left main, *n* (%)	82 (3.8)	17 (3.2)	0.524	32 (6.3)	13 (7.8)	0.508
LAD, *n* (%)	1175 (54.8)	278 (51.8)	0.207	298 (58.8)	90 (53.9)	0.268
LCx, *n* (%)	568 (26.5)	127 (23.6)	0.179	121 (23.9)	41 (24.6)	0.857
RCA, *n* (%)	997 (46.5)	253 (47.1)	0.799	223 (44.0)	99 (41.3)	0.589
ACC/AHA lesion type						
Type B1, *n* (%)	303 (14.1)	109 (20.3)	<0.001	72 (14.2)	16 (9.6)	0.145
Type B2, *n* (%)	720 (33.6)	133 (24.8)	<0.001	174 (34.3)	57 (34.1)	0.965
Type C, *n* (%)	983 (45.8)	224 (41.7)	0.090	238 (46.9)	83 (49.7)	0.536
Extent of CAD						
Single-vessel, *n* (%)	875 (40.8)	211 (39.3)	0.521	198 (39.1)	63 (37.7)	0.760
2-vessel, *n* (%)	671 (31.3)	172 (32.0)	0.743	148 (29.2)	52 (31.1)	0.633
≥3-vessel, *n* (%)	598 (27.9)	154 (28.7)	0.717	161 (31.8)	52 (31.1)	0.924
Pre-PCI TIMI 0/1, *n* (%)	1258 (58.7)	330 (61.5)	0.242	227 (44.8)	83 (49.7)	0.268
Type of stent						
ZES, *n* (%)	801 (37.4)	233 (43.4)	0.010	167 (32.9)	65 (38.9)	0.158
EES, *n* (%)	1092 (50.9)	245 (45.9)	0.030	274 (54.0)	79 (47.3)	0.153
BES, *n* (%)	249 (11.6)	52 (9.7)	0.222	64 (12.6)	18 (10.8)	0.587
Others, *n* (%)	43 (2.0)	12 (2.2)	0.738	8 (1.6)	7 (4.2)	0.065
IVUS, *n* (%)	401 (18.7)	85 (15.8)	0.133	89 (17.6)	30 (18.0)	0.907
OCT, *n* (%)	4 (0.2)	3 (0.6)	0.131	1 (0.2)	0	0.566
FFR, *n* (%)	5 (0.2)	3 (0.6)	0.216	2 (0.4)	1 (0.6)	0.575
Stent diameter (mm)	3.11 ± 0.42	3.11 ± 0.44	0.887	3.09 ± 0.43	3.11 ± 0.44	0.555
Stent length (mm)	27.8 ± 12.3	26.0 ± 11.1	0.002	29.0 ± 13.7	28.1 ± 14.4	0.497
Number of stents	1.54 ± 0.82	1.53 ± 0.82	0.648	1.60 ± 0.87	1.51 ± 0.76	0.196

Values are means ± SDs or numbers and percentages. The *p* values for continuous data were obtained from the unpaired *t*-test. The *p* values for categorical data were obtained from the chi-square or Fisher’s exact test. eGFR, estimated glomerular filtration rate; LVEF, left ventricular ejection fraction; BMI, body mass index; SBP, systolic blood pressure; DBP, diastolic blood pressure; CPR, cardiopulmonary resuscitation; STEMI, ST-segment elevation myocardial infarction; PCI, percutaneous coronary intervention; NSTEMI, non-STEMI; MI, myocardial infarction; CABG, coronary artery bypass graft; HF, heart failure; CVA, cerebrovascular accidents; CK-MB, creatine kinase myocardial band; NT-ProBNP, N-terminal pro-brain natriuretic peptide; CRP, C-reactive protein; HDL, high-density lipoprotein; LDL, low-density lipoprotein; ACEIs, angiotensin-converting enzyme inhibitors; ARBs, angiotensin receptor blockers; BBs, beta blockers; CCBs, calcium channel blockers; LAD, left anterior descending coronary artery; LCx, left circumflex coronary artery; RCA, right coronary artery; ACC/AHA, American College of Cardiology/American Heart Association; CAD, coronary artery disease; TIMI, thrombolysis in myocardial infarction; ZES, zotarolimus-eluting stent; EES, everolimus-eluting stent; BES, biolimus-eluting stent; IVUS, intravascular ultrasound; OCT, optical coherence tomography; FFR, fractional flow reserve.

**Table 5 jcm-10-03504-t005:** Hazard ratios for the 2-year major clinical outcomes in statin users.

	Hazard Ratio (95% CI) Unadjusted	*p* Value	Event Rates at 2 Years ^a^	Hazard Ratio (95% CI) Adjusted ^b^	*p* Value
**MACE**					
Group A1 vs.	-		5.2 %	-	-
Group A2	1.228 (1.059–1.425)	0.017	6.4 %	1.139 (0.969–1.339)	0.114
Group A3	2.015 (1.689–2.404)	<0.001	10.1 %	1.465 (1.183–1.813)	<0.001
Group A4	3.804 (2.991–4.837)	<0.001	18.5 %	2.082 (1.514–2.863)	<0.001
Group A2 vs. Group A3	1.641 (1.383–1.946)	<0.001		1.249 (1.027–1.520)	0.026
Group A2 vs. Group A4	3.096 (2.445–3.922)	<0.001		1.701 (1.263–2.290)	<0.001
Group A3 vs. Group A4	1.881 (1.458–2.427)	<0.001		1.439 (1.059–1.954)	0.020
**All-cause death**					
Group A1 vs.			0.9 %		-
Group A2	2.076 (1.494–2.886)	<0.001	1.8 %	1.937 (1.348–2.784)	<0.001
Group A3	5.454 (3.887–7.652)	<0.001	4.6 %	3.691 (2.452–5.554)	<0.001
Group A4	15.55 (10.71–22.56)	<0.001	12.9 %	5.068 (3.037–8.459)	<0.001
Group A2 vs. Group A3	2.625 (1.985–3.471)	<0.001		1.843 (1.342–2.531)	<0.001
Group A2 vs. Group A4	7.512 (5.457–10.34)	<0.001		3.160 (2.104–4.745)	<0.001
Group A3 vs. Group A4	2.853 (2.052–3.966)	<0.001		2.060 (1.396–3.039)	<0.001
**Cardiac death**					
Group A1 vs.			0.4 %		-
Group A2	2.364 (1.504–3.714)	<0.001	1.0 %	1.964 (1.215–3.177)	0.006
Group A3	6.001 (3.764–9.568)	<0.001	2.6 %	3.429 (1.993–5.898)	<0.001
Group A4	15.28 (9.047–25.81)	<0.001	6.8 %	4.512 (2.318–8.783)	<0.001
Group A2 vs. Group A3	2.537 (1.752–3.675)	<0.001		1.647 (1.087–2.495)	0.019
Group A2 vs. Group A4	6.476 (4.166–10.07)	<0.001		2.829 (1.674–4.781)	<0.001
Group A3 vs. Group A4	2.540 (1.610–4.008)	<0.001		2.040 (1.218–3.418)	0.007
**Recurrent MI**					
Group A1 vs.			1.6 %		-
Group A2	1.040 (0.785–1.377)	0.786	1.6 %	1.060 (0.778–1.444)	0.712
Group A3	1.738 (1.244–2.429)	0.001	2.8 %	1.070 (0.708–1.616)	0.750
Group A4	2.658 (1.607–4.395)	<0.001	4.3 %	1.486 (0.786–2.806)	0.223
Group A2 vs. Group A3	1.675 (1.675–2.340)	0.002		1.218 (0.821–1.807)	0.327
Group A2 vs. Group A4	2.556 (1.546–4.225)	<0.001		1.430 (0.762–2.682)	0.265
Group A3 vs. Group A4	1.531 (0.897–2.614)	0.118		1.192 (0.627–2.265)	0.593
**Any repeat revascularization**					
Group A1 vs.			3.4 %		
Group A2	1.092 (0.902–1.323)	0.367	3.6 %	1.036 (0.843–1.274)	0.735
Group A3	1.308 (1.013–1.689)	0.039	4.3 %	1.010 (0.743–1.372)	0.950
Group A4	1.582 (1.019–2.456)	0.041	5.2 %	1.161 (0.680–1.981)	0.585
Group A2 vs. Group A3	1.197 (0.929–1.542)	0.164		1.057 (0.791–1.413)	0.706
Group A2 vs. Group A4	1.442 (0.930–2.236)	0.102		1.053 (0.629–1.762)	0.845
Group A3 vs. Group A4	1.202 (0.751–1.924)	0.442		1.019 (0.587–1.771)	0.945

^a^ Event rates at 2 years were calculated by Kaplan–Meyer analysis. ^b^ Adjusted model included male, age, LVEF, BMI, cardiogenic shock, CPR on admission, Killip class III/IV, STEMI hypertension, diabetes mellitus, previous MI, PCI and CVA, current smoker, NT-ProBNP, blood glucose, total cholesterol, HDL cholesterol, ACEI, ARB, BB, LM (IRA and treated vessel), ACC/AHA type B2 lesion, single-vessel disease, ≥3-vessel disease, stent diameter, stent length and number of stents. Group A1, statin users and eGFR ≥ 90 mL/min/1.73 m^2^; Group A2, statin users and eGFR 60–89 mL/min/1.73 m^2^; Group A3, statin users and eGFR 30–59 mL/min/1.73 m^2^; Group A4, statin users and eGFR < 30 mL/min/1.73 m^2^; Group B1, statin nonusers and eGFR ≥ 90 mL/min/1.73 m^2^; Group B2, statin nonusers and eGFR 60–89 mL/min/1.73 m^2^; Group B3, statin nonusers and eGFR 30–59 mL/min/1.73 m^2^; Group B4, statin nonusers and eGFR < 30 mL/min/1.73 m^2^; eGFR, estimated glomerular filtration rate; CI, confidence interval; LVEF, left ventricular ejection fraction; BMI, body mass index; CPR, cardiopulmonary resuscitation; STEMI, ST-segment elevation myocardial infarction; MI, myocardial infarction; PCI, percutaneous coronary intervention; CVA, cerebrovascular accident; HDL, high-density lipoprotein; ACEI, angiotensin-converting enzyme inhibitor; ARB, angiotensin receptor blocker; BB, beta blocker; ACC/AHA, American College of Cardiology/American Heart Association.

**Table 6 jcm-10-03504-t006:** Clinical outcomes between statin users and nonusers at 2 years.

	Statin Users	Statin Nonusers					
Outcomes	Group A1 (*n* = 6847)	Group B1 (*n* = 889)	Log-Rank	Unadjusted		Adjusted ^a^	
				HR (95% CI)	*p* Value	HR (95% CI)	*p* Value
MACE	320 (5.2)	78 (9.0)	<0.001	1.821 (1.421–2.332)	<0.001	1.573 (1.181–2.096)	0.002
All-cause death	53 (0.9)	30 (3.4)	<0.001	4.235 (2.705–6.629)	<0.001	2.242 (1.261–3.984)	0.006
Cardiac death	27 (0.4)	20 (2.3)	<0.001	5.567 (3.121–9.929)	<0.001	2.956 (1.412–6.189)	0.004
Re-MI	97 (1.6)	19 (2.2)	0.138	1.448 (0.886–2.369)	0.140	1.578 (0.918–2.711)	0.099
Any revascularization	203 (3.4)	33 (4.0)	0.353	1.190 (0.824–1.720)	0.354	1.191 (0.796–1.783)	0.395
**Outcomes**	**Group A2** **(*n* = 6557)**	**Group B2** **(*n* = 1227)**	**Log-Rank**	**Unadjusted**		**Adjusted ^b^**	
				**HR (95% CI)**	***p* Value**	**HR (95% CI)**	***p* Value**
MACE	381 (6.4)	117 (9.7)	<0.001	1.573 (1.279–1.935)	<0.001	1.381 (1.092–1.747)	0.007
All-cause death	107 (1.8)	63 (5.2)	<0.001	3.031 (2.220–4.138)	<0.001	2.139 (1.471–3.110)	<0.001
Cardiac death	62 (1.0)	48 (4.0)	<0.001	4.023 (2.759–5.864)	<0.001	2.422 (1.536–3.819)	<0.001
Re-MI	98 (1.6)	19 (1.6)	0.980	1.006 (0.616–1.645)	0.980	1.199 (0.719–2.001)	0.487
Any revascularization	215 (3.6)	49 (4.2)	0.345	1.161 (0.851–1.584)	0.345	1.196 (0.856–1.671)	0.294
**Outcomes**	**Group A3** **(*n* = 2144)**	**Group B3** **(*n* = 537)**	**Log-Rank**	**Unadjusted**		**Adjusted ^c^**	
				**HR (95% CI)**	***p* Value**	**HR (95% CI)**	***p* Value**
MACE	201 (10.1)	90 (17.0)	<0.001	1.817 (1.417–2.330)	<0.001	1.732 (1.329–2.266)	<0.001
All-cause death	91 (4.6)	66 (12.4)	<0.001	2.978 (2.169–4.088)	<0.001	2.510 (1.780–3.541)	<0.001
Cardiac death	51 (2.6)	50 (9.5)	<0.001	4.034 (2.731–5.958)	<0.001	3.150 (2.069–4.795)	<0.001
Re-MI	53(2.8)	15 (3.1)	0.661	1.137 (0.641–2.017)	0.661	1.311 (0.735–2.409)	0.346
Any revascularization	83 (4.3)	29 (6.1)	0.111	1.407 (0.922–2.148)	0.113	1.532 (1.018–2.428)	0.086
**Outcomes**	**Group A4** **(*n* = 507)**	**Group B4** **(*n* = 167)**	**Log-Rank**	**Unadjusted**		**Adjusted ^d^**	
				**HR (95% CI)**	***p* Value**	**HR (95% CI)**	***p* Value**
MACE	84 (18.5)	52 (32.2)	<0.001	2.009 (1.422–2.840)	<0.001	1.949 (1.347–2.822)	<0.001
All-cause death	58 (12.9)	44 (27.3)	<0.001	2.493 (1.685–3.690)	<0.001	2.476 (1.629–3.755)	<0.001
Cardiac death	29 (6.8)	30 (19.0)	<0.001	3.378 (2.027–5.628)	<0.001	3.341 (1.975–5.706)	<0.001
Re-MI	18 (4.3)	6 (4.4)	0.901	1.060 (0.421–2.672)	0.901	1.065 (0.410–2.764)	0.898
Any revascularization	22 (5.2)	10 (7.9)	0.293	1.490 (0.705–3.146)	0.296	1.412 (0.639–3.118)	0.394

^a^ Adjusted by male, age, BMI, LVEF, cardiogenic shock, CPR on admission, STEMI, PCI within 24 h, hypertension, DM, NT-ProBNP, total cholesterol, LDL cholesterol, aspirin, clopidogrel, ticagrelor, prasugrel, ACEI, ARB, BB, ACC/AHA type B1 lesion, ≥3-vessel disease, ZES and stent length (Appendix A). ^b^ Adjusted by male, age, BMI, LVEF, cardiogenic shock, CPR on admission, STEMI, PCI within 24 h, hypertension, DM, dyslipidemia, NT-ProBNP, total cholesterol, LDL cholesterol, aspirin, clopidogrel, ticagrelor, prasugrel, ACEI, ARB, BB, ACC/AHA type C lesion, ≥3-vessel disease, ZES, BES, FFR and stent length (Appendix A). ^c^ Adjusted by male, age, BMI, SBP, DBP, LVEF, cardiogenic shock, CPR on admission, Killip class III/IV, STEMI, hypertension, DM, peak CK-MB, NT-ProBNP, total cholesterol, triglyceride, LDL cholesterol, aspirin, ticagrelor, ACEI, ARB, BB, ACC/AHA type B1/B2 lesions and ≥3-vessel disease (Appendix A). ^d^ Adjusted by male, age, BMI, LVEF, cardiogenic shock, CPR on admission, STEMI, hypertension, DM, NT-ProBNP, LDL cholesterol, ACE, ARB, BB and ≥3-vessel disease (Appendix A). Group A1, eGFR ≥ 90 mL/min/1.73 m^2^; Group A2, eGFR 60–89 mL/min/1.73 m^2^; Group A3, eGFR 30–59 mL/min/1.73 m^2^; Group A4, GFR < 30 mL/min/1.73 m^2^; Group B1, eGFR ≥ 90 mL/min/1.73 m^2^; Group B2, eGFR 60–89 mL/min/1.73 m^2^; Group B3, eGFR 30–59 mL/min/1.73 m^2^; Group B4, eGFR < 30 mL/min/1.73 m^2^; eGFR, estimated glomerular filtration rate; HR, hazard ratio; CI, confidence interval; MACE, major adverse cardiac events; Re-MI, recurrent myocardial infarction; BMI, body mass index; LVEF, left ventricular ejection fraction; CPR, cardiopulmonary resuscitation; STEMI, ST-segment elevation myocardial infarction; PCI, percutaneous coronary intervention; DM, diabetes mellitus; NT-ProBNP, N-terminal pro-brain natriuretic peptide; LDL, low-density lipoprotein; ACEI, angiotensin-converting enzyme inhibitors; ARB, angiotensin receptor blockers; BB, beta blocker; ACC/AHA, American College of Cardiology/American Heart Association; ZES, zotarolimus-eluting stent; BES, biolimus-eluting stent; FFR, fractional flow reserve; SBP, systolic blood pressure; DBP, diastolic blood pressure; CK-MB, creatine kinase myocardial band.

**Table 7 jcm-10-03504-t007:** Independent predictors for MACE and all-cause death in statin users.

Variables	MACE				All-Cause Death			
	Unadjusted		Adjusted		Unadjusted		Adjusted	
	HR (95% CI)	*p* Value	HR (95% CI)	*p* Value	HR (95% CI)	*p* Value	HR (95% CI)	*p* Value
Group A1 vs. Group A2	1.228 (1.059–1.425)	0.017	1.106 (0.940–1.301)	0.226	2.076 (1.494–2.886)	<0.001	1.146 (1.007–1.345)	0.040
Group A1 vs. Group A3	2.015 (1.689–2.404)	<0.001	1.314 (1.066–1.687)	0.012	5.454 (3.887–7.652)	<0.001	2.365 (1.544–3.625)	<0.001
Group A1 vs. Group A4	3.804 (2.991–4.837)	<0.001	1.779 (1.239–2.554)	0.002	15.55 (10.71–22.56)	<0.001	3.807 (2.151–6.736)	<0.001
Group A2 vs. Group A3	1.641 (1.383–1.946)	<0.001	1.342 (1.028–1.483)	0.040	2.625 (1.985–3.471)	<0.001	1.510 (1.085–2.100)	0.015
Group A2 vs. Group A4	3.096 (2.445–3.922)	<0.001	1.946 (1.522–2.415)	0.017	7.512 (5.457–10.34)	<0.001	2.175 (1.345–3.518)	0.002
Group A3 vs. Group A4	1.881 (1.458–2.427)	<0.001	1.521 (1.189–2.027)	0.034	2.853 (2.052–3.966)	<0.001	1.797 (1.162–2.779)	0.008
Male	1.325 (1.157–1.518)	<0.001	1.047 (0.888–1.235)	0.584	1.587 (1.255–2.006)	<0.001	1.146 (1.007–1.345)	0.040
Age, ≥65 years	1.577 (1.391–1.788)	<0.001	1.276 (1.094–1.488)	0.002	4.502 (3.430–5.910)	<0.001	3.985 (2.860–5.534)	<0.001
STEMI	1.300 (1.148–1.473)	<0.001	1.180 (1.028–1.355)	0.019	1.727 (1.377–2.166)	<0.001	1.425 (1.102–1.842)	0.007
LVEF, <40%	2.031 (1.724–2.393)	<0.001	1.491 (1.236–1.797)	<0.001	3.671 (2.865–4.703)	<0.001	2.088 (1.557–2.800)	<0.001
Cardiogenic shock	1.431 (1.074–1.907)	<0.001	1.369 (1.037–1.897)	0.005	2.304 (1.519–3.495)	<0.001	1.987 (1.269–3.231)	0.024
CPR on admission	2.680 (2.132–3.368)	<0.001	2.308 (1.787–2.982)	<0.001	3.979 (2.822–5.610)	<0.001	2.842 (1.900–4.252)	<0.001
Hypertension	1.498 (1.320–1.700)	<0.001	1.164 (1.007–1.345)	0.040	2.123 (1.676–2.678)	<0.001	1.390 (1.056–1.829)	0.019
Diabetes mellitus	1.791 (1.575–2.036)	<0.001	1.428 (1.234–1.653)	<0.001	2.391 (1.911–2.991)	<0.001	1.524 (1.172–1.980)	0.002
Previous heart failure	1.250 (0.690–2.266)	<0.001	1.613 (0.799–3.253)	0.182	2.590 (1.224–5.479)	0.013	1.058 (0.466–2.405)	0.892
Current smoker	1.309 (1.150–1.489)	<0.001	1.022 (0.873–1.197)	0.786	1.731 (1.360–2.202)	<0.001	1.214 (0.904–1.629)	0.197
NT-ProBNP	1.002 (0.999–1.004)	<0.001	1.003 (1.000–1.005)	<0.001	1.001 (0.998–1.002)	<0.001	1.002 (0.999–1.003)	<0.001
Total cholesterol	0.997 (0.995–0.998)	<0.001	1.001 (0.998–1.003)	0.637	0.994 (0.991–0.997)	<0.001	1.002 (0.998–1.007)	0.372
Triglyceride	0.999 (0.998–1.000)	<0.001	0.998 (0.997–0.999)	0.121	0.998 (0.996–0.999)	0.003	0.999 (0.997–1.001)	0.370
HDL cholesterol	0.990 (0.984–0.995)	<0.001	0.993 (0.988–1.001)	0.245	0.981 (0.976–0.997)	0.011	0.987 (0.975–0.999)	0.425
LDL cholesterol	0.998 (0.996–0.999)	<0.001	0.999 (0.997–1.000)	0.009	0.995 (0.992–0.999)	0.005	0.997 (0.992–1.001)	0.029
Aspirin	2.527 (1.491–4.283)	0.001	1.083 (0.553–2.121)	0.817	1.784 (1.356–2.351)	0.010	1.481 (0.661–3.318)	0.340
Ticagrelor	1.134 (0.874–1.471)	0.344	1.297 (0.972–1.730)	0.077	1.477 (0.877–2.486)	0.143	1.826 (0.993–3.360)	0.053
Prasugrel	1.344 (0.972–1.859)	0.074	1.256 (0.890–1.771)	0.194	3.224 (1.332–7.803)	0.009	2.356 (0.953–5.826)	0.063
ACEI	1.542 (1.360–1.747)	<0.001	1.369 (1.194–1.570)	<0.001	1.922 (1.532–2.411)	<0.001	1.613 (1.245–2.089)	<0.001
BB	1.360 (1.153–1.560)	<0.001	1.033 (0.856–1.245)	0.738	1.547 (1.167–2.051)	0.002	1.230 (0.902–1.679)	0.191
≥3-vessel disease	1.893 (1.656–2.164)	<0.001	1.572 (1.355–1.823)	<0.001	2.112 (1.671–2.668)	<0.001	1.374 (1.052–1.794)	0.020
Stent diameter <3.0 mm	1.189 (1.039–1.361)	0.012	1.038 (0.897–1.202)	0.614	1.075 (0.834–1.386)	0.575	1.333 (1.011–1.758)	0.041
Stent length ≥30 mm	1.250 (1.095–1.427)	0.001	1.096 (0.951–1.264)	0.206	1.499 (1.190–1.889)	0.001	1.263 (0.981–1.625)	0.070

MACE, major adverse cardiac events; HR, hazard ratio; CI, confidence interval; Group A1, statin users and eGFR ≥ 90 mL/min/1.73 m^2^; Group A2, statin users and eGFR 60–89 mL/min/1.73 m^2^; Group A3, statin users and eGFR 30–59 mL/min/1.73 m^2^; Group A4, statin users and eGFR < 30 mL/min/1.73 m^2^, eGFR, estimated glomerular filtration rate; STEMI, ST-elevation myocardial infarction; LVEF, left ventricular ejection fraction, CPR, cardiopulmonary resuscitation; NT-ProBNP, N-terminal pro-brain natriuretic peptide; HDL, high-density lipoprotein; LDL, low-density lipoprotein; ACEI, angiotensin-converting enzyme inhibitor; BB, beta blocker.

## Data Availability

Data are contained within the article or Appendix A.

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
