# Peer review of "Efficacy of Statin Treatment According to Baseline Renal Function in Korean Patients with Acute Myocardial Infarction Not Requiring Dialysis Undergoing Newer-Generation Drug-Eluting Stent Implantation"

_jcm, 2021, doi:10.3390/jcm10163504_

Round 1

Reviewer 1 Report

This is a two-year study of the efficacy of statin treatment according to baseline renal function in patients with acute myocardial infarction (AMI) not requiring dialysis undergoing newer-generation drug-eluting stent (DES) implantation. The take home message from the study is that with the newer-generation DES, although the statin treatment was effective in reducing mortality, this beneficial effect was diminished in accordance with the deterioration of baseline renal function. Though the data is disorganized and tables are hard to follow, the study could be of importance.

1) Does statin therapy decrease restenosis in patients with high eGFR.

2) Does the new generation DES reduced the rates of restenosis and repeat revascularization in base line eGFR or compromised patients with high eGFR.

3) Did the authors observed side effects of the drug-eluting stents such as thrombosis

4) Please state clearly in the abstract why statin treatment was not effective in reducing mortality in patients with renal disease.

Author Response

Response to Reviewer 1 Comments

Comments and Suggestions for Authors

This is a two-year study of the efficacy of statin treatment according to baseline renal function in patients with acute myocardial infarction (AMI) not requiring dialysis undergoing newer-generation drug-eluting stent (DES) implantation. The take home message from the study is that with the newer-generation DES, although the statin treatment was effective in reducing mortality, this beneficial effect was diminished in accordance with the deterioration of baseline renal function. Though the data is disorganized and tables are hard to follow, the study could be of importance.

Comments:

Point 1: Does statin therapy decrease restenosis in patients with high eGFR.

Response 1: We sincerely thank you for reviewer’s valuable comments and question. We think that this question is very important question. Unfortunately, until recently, limited data are available and this focus was not fully evaluated. Hence, definite answers for this question may be incomplete. Although statins are effective in lowering primary and secondary cardiovascular disease event rates in patients with mild to moderate chronic kidney disease (CKD) (Baigent C, Landray MJ, Eetith C, et al. The effects of lowering LDL cholesterol with simvastatin plus ezetimibe in patients with CKD (study of heart and renal protection): a randomized placebo-controlled trial. Lancet 2011;377:2181-2192; Washam JB, Herzog CA, Beitelshess AL, Cohren MG, et al. Pharmacotherapy in CKD patients presenting with acute coronary syndrome: a scientific statement from the American Heart Association. Circulation 2015;131;1123; Cholesterol Treatment Trialists Collaboration, Herrington WG, Emberson J, Mihaylova B, Blackwell L et al. Impact of renal function on the effects of LDL cholesterol lowering with statin-based regimens: a meta-analysis of individual participant data form 28 randomized trials, Lancet Diabetes Endocrionol 2016;4:829-839, cited in our paper reference [14]), patients with advanced CKD and end stage renal disease (ESRD) have been excluded from randomized clinical trials of statin therapy in ACS. Therefore, fewer trial data are available to evaluated treatment efficacy of statin in patients with advanced CKD. As mentioned in the “Discussion” section, Natusaki et al. [36] evaluated the therapeutic effect of statins on cardiovascular outcomes in patients with advanced CKD after PCI or coronary artery bypass graft. The cumulative incidences of any repeat revascularization of the severe CKD (eGFR <30mL/min/1.73m2) group was not significantly different between statin users and nonusers groups (p = 0.23). They suggested that patients with advanced CKD generally have advanced atherosclerosis, typically characterized by heavy calcification, and statin may no longer provide significant benefits in patients with end-stage vascular pathology. Similarly, in the Herrington et al. study [14], the cumulative incidences of any repeat revascularization of the severe CKD (eGFR <30mL/min/1.73m2) group was not significantly different between statin users and nonusers groups (relative risk: 0.78 [0.57-1.05]). Hence, we think that future studies specifically focused on the advanced CKD (eGFR <30mL/min/1.73m2) may help to clarify the benefit of statin treatment after PCI in this group.

We revised “Discussion” section as follows;

Before,

  1. Discussion

             The main findings of this retrospective observational study including patients with AMI who underwent successful PCI with newer-generation DES implantation were as follows: (1) regardless of the baseline renal function, individuals who underwent statin treatment had reduced rates of MACE, all-cause mortality, and CD than those in statin nonusers; (2) despite these beneficial effects of statin therapy, the MACE, all-cause death, and CD rates were significantly increased as the baseline eGFR decreased; (3) older age, STEMI, reduced LVEF, cardiogenic shock, CPR on admission, NT-ProBNP, LDL-cholesterol, ACEI levels, ≥ 3-vessel disease, and LM (IRA) were common independent predictors for both MACE and all-cause mortality in the statin user group.

             To date, in the current guidelines [7,8], despite an MVE reducing the benefit of statin therapy in predialysis patients, this beneficial effect of statin therapy was not distinguished according to renal function (e.g., CKD grade 3, 4, or 5), and there is no convincing evidence among patients on dialysis [10]. In our study, we only included patients with AMI who did not require dialysis. Additionally, we directly compared major clinical outcomes between the statin user and nonuser groups according to the baseline renal function to evaluate the presence or absence of benefit of statin treatment in these different renal function groups. In Table 3, in all four groups (eGFR ≥90, 60–89, 30–59, and <30 mL/min/1.73 m2), statin therapy significantly reduced the rates of MACE, all-cause mortality, and CD compared with those in statin nonusers. Moreover, the rates of MACE, all-cause mortality, and CD significantly increased as the baseline eGFR decreased in group A (Table 2). These findings could be related to the poorer baseline characteristics of the statin nonuser group (e.g., reduced LVEF, high numbers of patients with cardiogenic shock or CPR on admission, and high mean level of NT-ProBNP; Table S3 and S4) compared with statin users. However, our results were consistent with those of previous reports [15,27,28]. Palmer et al. [15] showed that statins reduced the all-cause mortality (relative risk [RR], 0.81; 95% CI, 0.74–1.88) and CD (RR, 0.78; 95% CI, 0.68–0.89) rates compared with placebo or no treatment in individuals not receiving dialysis. Sarnak et al. [10] also mentioned that the benefit of reducing MVE with statin-based therapy decreases as eGFR declines. Similarly, Herrington et al. [14] demonstrated that smaller relative effects of MVE were observed as eGFR declined (RR, 0.78; 99% CI, 0.75–0.82 for eGFR ≥60 mL/min/1.73 m2; RR, 0.76; 99% CI, 0.70–0.81 for eGFR 45–60 mL/min/1.73 m2; RR, 0.85; 99% CI, 0.75–0.96 for eGFR 30 to <45 mL/min/1.73 m2; RR, 0.85; 95% CI, 0.71–1.02 for eGFR <30 mL/min/1.73 m2). In group B, the MACE rates between groups B1 and B3 and between groups B2 and B3 and all-cause mortality and CD rates between groups B1 and B2 and between groups B2 and B3 were not significantly different (Table S5). However, after statin treatment (group A), the rates of MACE between groups A1 and A3 (aHR, 1.465; p <0.001) and groups A2 and A3 (aHR, 1.249; p = 0.026) and all-cause mortality and CD rates between groups A1 and A3 (aHR, 3.691; p < 0.001, aHR, 3.429; p < 0.001, respectively) and between groups A2 and A3 (aHR, 1.843; p < 0.001, aHR, 1.647; p = 0.019, respectively) were significantly different. These results could reflect the trend that if GFR is reduced, the relative beneficial effects of statins might be smaller in accordance with previous reports [10,14]. Although the precise mechanisms responsible for the pattern of diminished benefit of statin with lower renal function are not well-known, the peculiar characteristics of the patients with CKD could be related to this pattern [14]. Patients with CKD are often excluded from randomized trials that evaluate cardioprotective drugs, and the quality and coverage of evidence on which to guide decision making in this population is suboptimal [29]. This lack of evidence on optimal treatment strategies for such patients may result in worse outcomes. [30] Additionally, The cause of CD is influenced by misclassification of their atypical clinical presentation [31]. The difficulty of interpreting raised biomarkers of cardiac damage in CKD is a possible contributable factor [32]. As the GFR declines, vascular calcification increases, and the calcification of the intima and media of large vessels in CKD is associated with all-cause death and cardiovascular mortality [10,34]. These cardiovascular changes in CKD are related to traditional (e.g., diabetes and hypertension) and nontraditional CKD-related cardiovascular disease risk factors (e.g., mineral and bone disease abnormalities, inflammation, and oxidative stress) [10]. Because there is geographical variation in the prevalence of DM, the absolute magnitude of beneficial effects of statin therapy can vary regionally [35].     

             In our study, the re-MI and any repeat revascularization rates were not significantly different between the statin user and nonuser groups. Similar results were reported by Natsuaki et al. [36]. Among 14706 patients who underwent PCI [36], the number of patients with AMI was approximately 30%. During a median follow-up of 956 days, the re-MI and any repeat revascularization rates were not significantly different between the statin user and nonuser groups according to the three different renal function groups (eGFR ≥60, ≤30 to <60, and <30 mL/min/1.73 m2). Another randomized study [37] failed to show the effects of statin therapy in decreasing restenosis. Although the study population was not confined to individual with AMI or CKD, according to the Cholesterol Treatment Trialists’ (CTT) Collaboration report, intensive statin therapy reduced the coronary revascularization rate about 19% (95% CI, 11-18; p <0.0001) [38]. Walter et al. [39] have found that patients receiving prolonged statin treatment developed lower in-stent restenosis rates in comparison with nonreceivers (25% vs. 38%). Therefore, our results showing similar Re-MI and any repeat revascularization rates between statin users and nonusers could be related with low number of enrolled patients in groups A4, B1, B3, and B4, and relatively low incidences of these events compared with previous studies [17,30]. Interestingly, the number of patients with NSTEI was increased as their renal function was deteriorated (Table 1 and S2). Although, the precise underlying mechanisms of this phenomenon are not well-known, some suggestions include that plaque erosion may be a more predominant in the CKD group, in those patients tend to be older and have more established atherosclerosis. Whereas the incidence of plaque rupture may be more common in younger non-CKD patients where less mature plaques that are more vulnerable to rupture [40,41]. In both the statin user and nonuser groups, reduced LVEF, cardiogenic shock, CPR on admission, and NT-ProBNP, and LDL-cholesterol levels were common independent predictors for both MACE and all-cause mortality. These variables are well-known unfavorable risk factors for mortality in patients with AMI [7,8].

             Because the study population of previous studies [14-17,27,28,36] regarding the long-term effects of statin treatment on major adverse events in patients with CKD was not confined to individuals with AMI and who received newer-generation DES, we investigated long-term major clinical outcomes of statin therapy confined to those patients to reflect current real-world practice. Moreover, as mentioned [29], evidences for optimal treatment strategies in patients with CKD are not abundant. More than 50 high-volume universities or community hospitals in South Korea participated in this study, but the study population was insufficient to provide meaningful results. Despite this weak point, we believe that our results could provide helpful information to interventional cardiologists in terms of current real-world information showing long-term effects of statin treatment according to the different renal function groups.

             This study had other limitations. First, there may have been some underreporting and/or missed data and selection bias because this was a nonrandomized study. Second, although micaroalbuminuria is an early marker of chronic renal damage and a risk factor of cardiovascular disease [42], there was likely some misclassification of study groups due to the lack of information concerning the total amount of proteinuria and the presence or absence of microalbuminuria. Third, the estimation of renal function was based on a single measurement of eGFR at the time of presentation to the hospital. However, there may be a possibility that eGFR may have worsened during the follow-up period. Unfortunately, we could not provide follow-up eGFR values because of the limitation of this registry data. Fourth, according to the current guidelines [7], the treatment goal is an LDL-cholesterol level <1.8 mmol/L (<70 mg/dL) or at least 50% reduction in LDL-cholesterol if the baseline LDL-cholesterol level is 1.8–3.5 mmol/L. However, information regarding the follow-up levels of blood LDL-cholesterol was incomplete in our registry data. This is a major shortcoming of this study and may be an important bias. Fifth, because the registry data did not include detailed or complete data on prescription doses, long-term adherence, discontinuation, and drug-related adverse events, we could not provide this information during the follow-up period, which could have caused bias. Sixth, despite multivariable analyses, the variables that were not included in the data registry might have affected the study outcome. Seventh, because statins have a longer duration of use, the 2-year follow-up period in this study was relatively short to estimate the long-term clinical outcomes. Finally, because this retrospective study enrolled patients who underwent PCI between January 2006 and June 2015, this broad timeframe could have affected the clinical outcomes.

After,

  1. Discussion

The main findings of this retrospective observational study including patients with AMI who underwent successful PCI with newer-generation DES implantation were as follows: (1) regardless of the baseline renal function, individuals who underwent statin treatment had reduced rates of MACE, all-cause mortality, and CD than those in statin nonusers; (2) despite these beneficial effects of statin therapy, the MACE, all-cause death, and CD rates were significantly increased as the baseline eGFR decreased; (3) older age, STEMI, reduced LVEF, cardiogenic shock, CPR on admission, NT-ProBNP, LDL-cholesterol, ACEI levels, ≥ 3-vessel disease, and LM (IRA) were common independent predictors for both MACE and all-cause mortality in the statin user group.

             To date, in the current guidelines [7,8], despite an MVE reducing the benefit of statin therapy in predialysis patients, this beneficial effect of statin therapy was not distinguished according to renal function (e.g., CKD grade 3, 4, or 5), and there is no convincing evidence among patients on dialysis [10]. In our study, we only included patients with AMI who did not require dialysis. Additionally, we directly compared major clinical outcomes between the statin user and nonuser groups according to the baseline renal function to evaluate the presence or absence of benefit of statin treatment in these different renal function groups. In Table 3, in all four groups (eGFR ≥90, 60–89, 30–59, and <30 mL/min/1.73 m2), statin therapy significantly reduced the rates of MACE, all-cause mortality, and CD compared with those in statin nonusers. Moreover, the rates of MACE, all-cause mortality, and CD significantly increased as the baseline eGFR decreased in group A (Table 2). These findings could be related to the poorer baseline characteristics of the statin nonuser group (e.g., reduced LVEF, high numbers of patients with cardiogenic shock or CPR on admission, and high mean level of NT-ProBNP; Table S3 and S4) compared with statin users. However, our results were consistent with those of previous reports [15,27,28]. Palmer et al. [15] showed that statins reduced the all-cause mortality (relative risk [RR], 0.81; 95% CI, 0.74–1.88) and CD (RR, 0.78; 95% CI, 0.68–0.89) rates compared with placebo or no treatment in individuals not receiving dialysis. Sarnak et al. [10] also mentioned that the benefit of reducing MVE with statin-based therapy decreases as eGFR declines. Similarly, Herrington et al. [14] demonstrated that smaller relative effects of MVE were observed as eGFR declined (RR, 0.78; 99% CI, 0.75–0.82 for eGFR ≥60 mL/min/1.73 m2; RR, 0.76; 99% CI, 0.70–0.81 for eGFR 45–60 mL/min/1.73 m2; RR, 0.85; 99% CI, 0.75–0.96 for eGFR 30 to <45 mL/min/1.73 m2; RR, 0.85; 95% CI, 0.71–1.02 for eGFR <30 mL/min/1.73 m2). In group B, the MACE rates between groups B1 and B3 and between groups B2 and B3 and all-cause mortality and CD rates between groups B1 and B2 and between groups B2 and B3 were not significantly different (Table S5). However, after statin treatment (group A), the rates of MACE between groups A1 and A3 (aHR, 1.465; p <0.001) and groups A2 and A3 (aHR, 1.249; p = 0.026) and all-cause mortality and CD rates between groups A1 and A3 (aHR, 3.691; p < 0.001, aHR, 3.429; p < 0.001, respectively) and between groups A2 and A3 (aHR, 1.843; p < 0.001, aHR, 1.647; p = 0.019, respectively) were significantly different. These results could reflect the trend that if GFR is reduced, the relative beneficial effects of statins might be smaller in accordance with previous reports [10,14]. Although the precise mechanisms responsible for the pattern of diminished benefit of statin with lower renal function are not well-known, the peculiar characteristics of the patients with CKD could be related to this pattern [14]. Patients with CKD are often excluded from randomized trials that evaluate cardioprotective drugs, and the quality and coverage of evidence on which to guide decision making in this population is suboptimal [29]. This lack of evidence on optimal treatment strategies for such patients may result in worse outcomes. [30] Additionally, The cause of CD is influenced by misclassification of their atypical clinical presentation [31]. The difficulty of interpreting raised biomarkers of cardiac damage in CKD is a possible contributable factor [32]. As the GFR declines, vascular calcification increases, and the calcification of the intima and media of large vessels in CKD is associated with all-cause death and cardiovascular mortality [10,34]. These cardiovascular changes in CKD are related to traditional (e.g., diabetes and hypertension) and nontraditional CKD-related cardiovascular disease risk factors (e.g., mineral and bone disease abnormalities, inflammation, and oxidative stress) [10]. Because there is geographical variation in the prevalence of DM, the absolute magnitude of beneficial effects of statin therapy can vary regionally [35].     

             In our study, the re-MI and any repeat revascularization rates were not significantly different between the statin user and nonuser groups. Similar results were reported by Natsuaki et al. [36]. Among 14706 patients who underwent PCI [36], the number of patients with AMI was approximately 30%. During a median follow-up of 956 days, the re-MI and any repeat revascularization rates were not significantly different between the statin user and nonuser groups according to the three different renal function groups (eGFR ≥60, ≤30 to <60, and <30 mL/min/1.73 m2). They [36] suggested that patients with advanced CKD (eGFR <30mL/min/1.73m2) generally have advanced atherosclerosis, typically characterized by heavy calcification, and statin may no longer provide significant benefits in patients with end-stage vascular pathology. Another randomized study [37] failed to show the effects of statin therapy in decreasing restenosis. Although the study population was not confined to individual with AMI or CKD, according to the Cholesterol Treatment Trialists’ (CTT) Collaboration report, intensive statin therapy reduced the coronary revascularization rate about 19% (95% CI, 11-18; p <0.0001) [38]. Walter et al. [39] have found that patients receiving prolonged statin treatment developed lower in-stent restenosis rates in comparison with nonreceivers (25% vs. 38%). Therefore, our results showing similar Re-MI and any repeat revascularization rates between statin users and nonusers could be related with low number of enrolled patients in groups A4, B1, B3, and B4, and relatively low incidences of these events compared with previous studies [17,30]. However, we think that future studies specifically focused on the advanced CKD may help to clarify the benefit of statin treatment after PCI in this group. Interestingly, the number of patients with NSTEI was increased as their renal function was deteriorated (Table 1 and S2). Although, the precise underlying mechanisms of this phenomenon are not well-known, some suggestions include that plaque erosion may be a more predominant in the CKD group, in those patients tend to be older and have more established atherosclerosis. Whereas the incidence of plaque rupture may be more common in younger non-CKD patients where less mature plaques that are more vulnerable to rupture [40,41]. In both the statin user and nonuser groups, reduced LVEF, cardiogenic shock, CPR on admission, and NT-ProBNP, and LDL-cholesterol levels were common independent predictors for both MACE and all-cause mortality. These variables are well-known unfavorable risk factors for mortality in patients with AMI [7,8].

Because the study population of previous studies [14-17,27,28,36] regarding the long-term effects of statin treatment on major adverse events in patients with CKD was not confined to individuals with AMI and who received newer-generation DES, we investigated long-term major clinical outcomes of statin therapy confined to those patients to reflect current real-world practice. Moreover, as mentioned [29], evidences for optimal treatment strategies in patients with CKD are not abundant. More than 50 high-volume universities or community hospitals in South Korea participated in this study, but the study population was insufficient to provide meaningful results. Despite this weak point, we believe that our results could provide helpful information to interventional cardiologists in terms of current real-world information showing long-term effects of statin treatment according to the different renal function groups.

This study had other limitations. First, there may have been some underreporting and/or missed data and selection bias because this was a nonrandomized study. Second, although micaroalbuminuria is an early marker of chronic renal damage and a risk factor of cardiovascular disease [42], there was likely some misclassification of study groups due to the lack of information concerning the total amount of proteinuria and the presence or absence of microalbuminuria. Third, the estimation of renal function was based on a single measurement of eGFR at the time of presentation to the hospital. However, there may be a possibility that eGFR may have worsened during the follow-up period. Unfortunately, we could not provide follow-up eGFR values because of the limitation of this registry data. Fourth, according to the current guidelines [7], the treatment goal is an LDL-cholesterol level <1.8 mmol/L (<70 mg/dL) or at least 50% reduction in LDL-cholesterol if the baseline LDL-cholesterol level is 1.8–3.5 mmol/L. However, information regarding the follow-up levels of blood LDL-cholesterol was incomplete in our registry data. This is a major shortcoming of this study and may be an important bias. Fifth, because the registry data did not include detailed or complete data on prescription doses, long-term adherence, discontinuation, and drug-related adverse events, we could not provide this information during the follow-up period, which could have caused bias. Sixth, despite multivariable analyses, the variables that were not included in the data registry might have affected the study outcome. Seventh, because statins have a longer duration of use, the 2-year follow-up period in this study was relatively short to estimate the long-term clinical outcomes. Finally, because this retrospective study enrolled patients who underwent PCI between January 2006 and June 2015, this broad timeframe could have affected the clinical outcomes.

Point 2: Does the new generation DES reduced the rates of restenosis and repeat revascularization in base line eGFR or compromised patients with high eGFR.

Response 2: We sincerely thank you for reviewer’s valuable question. The main aim of our study is to provide helpful information to interventional cardiologists in terms of current real-world information showing long-term effects of statin treatment according to the different renal function groups. Recent meta-analysis data (Rimi G, Gritti V, Galiffa VA, et al. Drug eluting stents are superior to bare metal stents to reduce clinical outcome and stent-related complications in CKD patients, a systematic review, meta-analysis and network meta-analysis. J Interv Cardiol. 2018;31:319-329) showed that the TLR/TVR (RR, 0.69; 95% CI, 0.57-0.84) were significantly reduced with DES compared with bare-metal stent (BMS). Additionally, the use of second-generation DES was associated with relative 27% reduction in TLR/TVR compared with first-generation DES. They evaluated CKD patients, and CKD was defined as eGFR <60mL/min/1.73m2. Other study meta-analysis data (Volodarskiy A, Kumar S, Pracon R, et al. (Drug-Eluting vs Bare-Metal Stents in Patients With Chronic Kidney Disease and Coronary Artery Disease: Insights From a Systematic Review and Meta-Analysis. J invasive Cardiol. 2018;30:10-17) showed that DES was associated with lower TVR (RR, 0.61; 95% CI, 0.47-0.80) when compared with BMS. Unfortunately, reports concerning beneficial effect of new-generation DES on reducing revascularization according to the grade of eGFR or confined to the patients with high eGFR are insufficient.

Therefore, we revised “Discussion” section once more as follows;

Before,

  1. Discussion

The main findings of this retrospective observational study including patients with AMI who underwent successful PCI with newer-generation DES implantation were as follows: (1) regardless of the baseline renal function, individuals who underwent statin treatment had reduced rates of MACE, all-cause mortality, and CD than those in statin nonusers; (2) despite these beneficial effects of statin therapy, the MACE, all-cause death, and CD rates were significantly increased as the baseline eGFR decreased; (3) older age, STEMI, reduced LVEF, cardiogenic shock, CPR on admission, NT-ProBNP, LDL-cholesterol, ACEI levels, ≥ 3-vessel disease, and LM (IRA) were common independent predictors for both MACE and all-cause mortality in the statin user group.

             To date, in the current guidelines [7,8], despite an MVE reducing the benefit of statin therapy in predialysis patients, this beneficial effect of statin therapy was not distinguished according to renal function (e.g., CKD grade 3, 4, or 5), and there is no convincing evidence among patients on dialysis [10]. In our study, we only included patients with AMI who did not require dialysis. Additionally, we directly compared major clinical outcomes between the statin user and nonuser groups according to the baseline renal function to evaluate the presence or absence of benefit of statin treatment in these different renal function groups. In Table 3, in all four groups (eGFR ≥90, 60–89, 30–59, and <30 mL/min/1.73 m2), statin therapy significantly reduced the rates of MACE, all-cause mortality, and CD compared with those in statin nonusers. Moreover, the rates of MACE, all-cause mortality, and CD significantly increased as the baseline eGFR decreased in group A (Table 2). These findings could be related to the poorer baseline characteristics of the statin nonuser group (e.g., reduced LVEF, high numbers of patients with cardiogenic shock or CPR on admission, and high mean level of NT-ProBNP; Table S3 and S4) compared with statin users. However, our results were consistent with those of previous reports [15,27,28]. Palmer et al. [15] showed that statins reduced the all-cause mortality (relative risk [RR], 0.81; 95% CI, 0.74–1.88) and CD (RR, 0.78; 95% CI, 0.68–0.89) rates compared with placebo or no treatment in individuals not receiving dialysis. Sarnak et al. [10] also mentioned that the benefit of reducing MVE with statin-based therapy decreases as eGFR declines. Similarly, Herrington et al. [14] demonstrated that smaller relative effects of MVE were observed as eGFR declined (RR, 0.78; 99% CI, 0.75–0.82 for eGFR ≥60 mL/min/1.73 m2; RR, 0.76; 99% CI, 0.70–0.81 for eGFR 45–60 mL/min/1.73 m2; RR, 0.85; 99% CI, 0.75–0.96 for eGFR 30 to <45 mL/min/1.73 m2; RR, 0.85; 95% CI, 0.71–1.02 for eGFR <30 mL/min/1.73 m2). In group B, the MACE rates between groups B1 and B3 and between groups B2 and B3 and all-cause mortality and CD rates between groups B1 and B2 and between groups B2 and B3 were not significantly different (Table S5). However, after statin treatment (group A), the rates of MACE between groups A1 and A3 (aHR, 1.465; p <0.001) and groups A2 and A3 (aHR, 1.249; p = 0.026) and all-cause mortality and CD rates between groups A1 and A3 (aHR, 3.691; p < 0.001, aHR, 3.429; p < 0.001, respectively) and between groups A2 and A3 (aHR, 1.843; p < 0.001, aHR, 1.647; p = 0.019, respectively) were significantly different. These results could reflect the trend that if GFR is reduced, the relative beneficial effects of statins might be smaller in accordance with previous reports [10,14]. Although the precise mechanisms responsible for the pattern of diminished benefit of statin with lower renal function are not well-known, the peculiar characteristics of the patients with CKD could be related to this pattern [14]. Patients with CKD are often excluded from randomized trials that evaluate cardioprotective drugs, and the quality and coverage of evidence on which to guide decision making in this population is suboptimal [29]. This lack of evidence on optimal treatment strategies for such patients may result in worse outcomes. [30] Additionally, The cause of CD is influenced by misclassification of their atypical clinical presentation [31]. The difficulty of interpreting raised biomarkers of cardiac damage in CKD is a possible contributable factor [32]. As the GFR declines, vascular calcification increases, and the calcification of the intima and media of large vessels in CKD is associated with all-cause death and cardiovascular mortality [10,34]. These cardiovascular changes in CKD are related to traditional (e.g., diabetes and hypertension) and nontraditional CKD-related cardiovascular disease risk factors (e.g., mineral and bone disease abnormalities, inflammation, and oxidative stress) [10]. Because there is geographical variation in the prevalence of DM, the absolute magnitude of beneficial effects of statin therapy can vary regionally [35].     

             In our study, the re-MI and any repeat revascularization rates were not significantly different between the statin user and nonuser groups. Similar results were reported by Natsuaki et al. [36]. Among 14706 patients who underwent PCI [36], the number of patients with AMI was approximately 30%. During a median follow-up of 956 days, the re-MI and any repeat revascularization rates were not significantly different between the statin user and nonuser groups according to the three different renal function groups (eGFR ≥60, ≤30 to <60, and <30 mL/min/1.73 m2). They [36] suggested that patients with advanced CKD (eGFR <30mL/min/1.73m2) generally have advanced atherosclerosis, typically characterized by heavy calcification, and statin may no longer provide significant benefits in patients with end-stage vascular pathology. Another randomized study [37] failed to show the effects of statin therapy in decreasing restenosis. Although the study population was not confined to individual with AMI or CKD, according to the Cholesterol Treatment Trialists’ (CTT) Collaboration report, intensive statin therapy reduced the coronary revascularization rate about 19% (95% CI, 11-18; p <0.0001) [38]. Walter et al. [39] have found that patients receiving prolonged statin treatment developed lower in-stent restenosis rates in comparison with nonreceivers (25% vs. 38%). Therefore, our results showing similar Re-MI and any repeat revascularization rates between statin users and nonusers could be related with low number of enrolled patients in groups A4, B1, B3, and B4, and relatively low incidences of these events compared with previous studies [17,30]. However, we think that future studies specifically focused on the advanced CKD may help to clarify the benefit of statin treatment after PCI in this group. Interestingly, the number of patients with NSTEI was increased as their renal function was deteriorated (Table 1 and S2). Although, the precise underlying mechanisms of this phenomenon are not well-known, some suggestions include that plaque erosion may be a more predominant in the CKD group, in those patients tend to be older and have more established atherosclerosis. Whereas the incidence of plaque rupture may be more common in younger non-CKD patients where less mature plaques that are more vulnerable to rupture [40,41]. In both the statin user and nonuser groups, reduced LVEF, cardiogenic shock, CPR on admission, and NT-ProBNP, and LDL-cholesterol levels were common independent predictors for both MACE and all-cause mortality. These variables are well-known unfavorable risk factors for mortality in patients with AMI [7,8].

Because the study population of previous studies [14-17,27,28,36] regarding the long-term effects of statin treatment on major adverse events in patients with CKD was not confined to individuals with AMI and who received newer-generation DES, we investigated long-term major clinical outcomes of statin therapy confined to those patients to reflect current real-world practice. Moreover, as mentioned [29], evidences for optimal treatment strategies in patients with CKD are not abundant. More than 50 high-volume universities or community hospitals in South Korea participated in this study, but the study population was insufficient to provide meaningful results. Despite this weak point, we believe that our results could provide helpful information to interventional cardiologists in terms of current real-world information showing long-term effects of statin treatment according to the different renal function groups.

This study had other limitations. First, there may have been some underreporting and/or missed data and selection bias because this was a nonrandomized study. Second, although micaroalbuminuria is an early marker of chronic renal damage and a risk factor of cardiovascular disease [42], there was likely some misclassification of study groups due to the lack of information concerning the total amount of proteinuria and the presence or absence of microalbuminuria. Third, the estimation of renal function was based on a single measurement of eGFR at the time of presentation to the hospital. However, there may be a possibility that eGFR may have worsened during the follow-up period. Unfortunately, we could not provide follow-up eGFR values because of the limitation of this registry data. Fourth, according to the current guidelines [7], the treatment goal is an LDL-cholesterol level <1.8 mmol/L (<70 mg/dL) or at least 50% reduction in LDL-cholesterol if the baseline LDL-cholesterol level is 1.8–3.5 mmol/L. However, information regarding the follow-up levels of blood LDL-cholesterol was incomplete in our registry data. This is a major shortcoming of this study and may be an important bias. Fifth, because the registry data did not include detailed or complete data on prescription doses, long-term adherence, discontinuation, and drug-related adverse events, we could not provide this information during the follow-up period, which could have caused bias. Sixth, despite multivariable analyses, the variables that were not included in the data registry might have affected the study outcome. Seventh, because statins have a longer duration of use, the 2-year follow-up period in this study was relatively short to estimate the long-term clinical outcomes. Finally, because this retrospective study enrolled patients who underwent PCI between January 2006 and June 2015, this broad timeframe could have affected the clinical outcomes.

After,

  1. Discussion

The main findings of this retrospective observational study including patients with AMI who underwent successful PCI with newer-generation DES implantation were as follows: (1) regardless of the baseline renal function, individuals who underwent statin treatment had reduced rates of MACE, all-cause mortality, and CD than those in statin nonusers; (2) despite these beneficial effects of statin therapy, the MACE, all-cause death, and CD rates were significantly increased as the baseline eGFR decreased; (3) older age, STEMI, reduced LVEF, cardiogenic shock, CPR on admission, NT-ProBNP, LDL-cholesterol, ACEI levels, ≥ 3-vessel disease, and LM (IRA) were common independent predictors for both MACE and all-cause mortality in the statin user group.

             To date, in the current guidelines [7,8], despite an MVE reducing the benefit of statin therapy in predialysis patients, this beneficial effect of statin therapy was not distinguished according to renal function (e.g., CKD grade 3, 4, or 5), and there is no convincing evidence among patients on dialysis [10]. In our study, we only included patients with AMI who did not require dialysis. Additionally, we directly compared major clinical outcomes between the statin user and nonuser groups according to the baseline renal function to evaluate the presence or absence of benefit of statin treatment in these different renal function groups. In Table 3, in all four groups (eGFR ≥90, 60–89, 30–59, and <30 mL/min/1.73 m2), statin therapy significantly reduced the rates of MACE, all-cause mortality, and CD compared with those in statin nonusers. Moreover, the rates of MACE, all-cause mortality, and CD significantly increased as the baseline eGFR decreased in group A (Table 2). These findings could be related to the poorer baseline characteristics of the statin nonuser group (e.g., reduced LVEF, high numbers of patients with cardiogenic shock or CPR on admission, and high mean level of NT-ProBNP; Table S3 and S4) compared with statin users. However, our results were consistent with those of previous reports [15,27,28]. Palmer et al. [15] showed that statins reduced the all-cause mortality (relative risk [RR], 0.81; 95% CI, 0.74–1.88) and CD (RR, 0.78; 95% CI, 0.68–0.89) rates compared with placebo or no treatment in individuals not receiving dialysis. Sarnak et al. [10] also mentioned that the benefit of reducing MVE with statin-based therapy decreases as eGFR declines. Similarly, Herrington et al. [14] demonstrated that smaller relative effects of MVE were observed as eGFR declined (RR, 0.78; 99% CI, 0.75–0.82 for eGFR ≥60 mL/min/1.73 m2; RR, 0.76; 99% CI, 0.70–0.81 for eGFR 45–60 mL/min/1.73 m2; RR, 0.85; 99% CI, 0.75–0.96 for eGFR 30 to <45 mL/min/1.73 m2; RR, 0.85; 95% CI, 0.71–1.02 for eGFR <30 mL/min/1.73 m2). In group B, the MACE rates between groups B1 and B3 and between groups B2 and B3 and all-cause mortality and CD rates between groups B1 and B2 and between groups B2 and B3 were not significantly different (Table S5). However, after statin treatment (group A), the rates of MACE between groups A1 and A3 (aHR, 1.465; p <0.001) and groups A2 and A3 (aHR, 1.249; p = 0.026) and all-cause mortality and CD rates between groups A1 and A3 (aHR, 3.691; p < 0.001, aHR, 3.429; p < 0.001, respectively) and between groups A2 and A3 (aHR, 1.843; p < 0.001, aHR, 1.647; p = 0.019, respectively) were significantly different. These results could reflect the trend that if GFR is reduced, the relative beneficial effects of statins might be smaller in accordance with previous reports. [10,14]. Although the precise mechanisms responsible for the pattern of diminished benefit of statin with lower renal function are not well-known, the peculiar characteristics of the patients with CKD could be related to this pattern [14]. Patients with CKD are often excluded from randomized trials that evaluate cardioprotective drugs, and the quality and coverage of evidence on which to guide decision making in this population is suboptimal [29]. This lack of evidence on optimal treatment strategies for such patients may result in worse outcomes. [30] Additionally, The cause of CD is influenced by misclassification of their atypical clinical presentation [31]. The difficulty of interpreting raised biomarkers of cardiac damage in CKD is a possible contributable factor [32]. As the GFR declines, vascular calcification increases, and the calcification of the intima and media of large vessels in CKD is associated with all-cause death and cardiovascular mortality [10,34]. These cardiovascular changes in CKD are related to traditional (e.g., diabetes and hypertension) and nontraditional CKD-related cardiovascular disease risk factors (e.g., mineral and bone disease abnormalities, inflammation, and oxidative stress) [10]. Because there is geographical variation in the prevalence of DM, the absolute magnitude of beneficial effects of statin therapy can vary regionally [35].     

             In our study, the re-MI and any repeat revascularization rates were not significantly different between the statin user and nonuser groups. Similar results were reported by Natsuaki et al. [36]. Among 14706 patients who underwent PCI [36], the number of patients with AMI was approximately 30%. During a median follow-up of 956 days, the re-MI and any repeat revascularization rates were not significantly different between the statin user and nonuser groups according to the three different renal function groups (eGFR ≥60, ≤30 to <60, and <30 mL/min/1.73 m2). They [36] suggested that patients with advanced CKD (eGFR <30mL/min/1.73m2) generally have advanced atherosclerosis, typically characterized by heavy calcification, and statin may no longer provide significant benefits in patients with end-stage vascular pathology. Another randomized study [37] failed to show the effects of statin therapy in decreasing restenosis. Although the study population was not confined to individual with AMI or CKD, according to the Cholesterol Treatment Trialists’ (CTT) Collaboration report, intensive statin therapy reduced the coronary revascularization rate about 19% (95% CI, 11-18; p <0.0001) [38]. Walter et al. [39] have found that patients receiving prolonged statin treatment developed lower in-stent restenosis rates in comparison with nonreceivers (25% vs. 38%). Therefore, our results showing similar Re-MI and any repeat revascularization rates between statin users and nonusers could be related with low number of enrolled patients in groups A4, B1, B3, and B4, and relatively low incidences of these events compared with previous studies [17,30]. According to a recent meta-analysis data that [40] evaluated CKD patients, and CKD was defined as eGFR <60mL/min/1.73m2, showed that the TLR/TVR (RR, 0.69; 95% CI, 0.57-0.84) were significantly reduced with DES compared with bare-metal stent (BMS). Additionally, the use of second-generation DES was associated with relative 27% reduction in TLR/TVR compared with first-generation DES. Other study meta-analysis data [41] showed that DES was associated with lower TVR (RR, 0.61; 95% CI, 0.47-0.80) when compared with BMS in patients with CKD. However, we think that future studies specifically focused on the advanced CKD may help to clarify the benefit of statin treatment after PCI in this group. Interestingly, the number of patients with NSTEI was increased as their renal function was deteriorated (Table 1 and S2). Although, the precise underlying mechanisms of this phenomenon are not well-known, some suggestions include that plaque erosion may be a more predominant in the CKD group, in those patients tend to be older and have more established atherosclerosis. Whereas the incidence of plaque rupture may be more common in younger non-CKD patients where less mature plaques that are more vulnerable to rupture [42,43]. In both the statin user and nonuser groups, reduced LVEF, cardiogenic shock, CPR on admission, and NT-ProBNP, and LDL-cholesterol levels were common independent predictors for both MACE and all-cause mortality. These variables are well-known unfavorable risk factors for mortality in patients with AMI [7,8].

Because the study population of previous studies [14-17,27,28,36] regarding the long-term effects of statin treatment on major adverse events in patients with CKD was not confined to individuals with AMI and who received newer-generation DES, we investigated long-term major clinical outcomes of statin therapy confined to those patients to reflect current real-world practice. Moreover, as mentioned [29], evidences for optimal treatment strategies in patients with CKD are not abundant. More than 50 high-volume universities or community hospitals in South Korea participated in this study, but the study population was insufficient to provide meaningful results. Despite this weak point, we believe that our results could provide helpful information to interventional cardiologists in terms of current real-world information showing long-term effects of statin treatment according to the different renal function groups.

This study had other limitations. First, there may have been some underreporting and/or missed data and selection bias because this was a nonrandomized study. Second, although micaroalbuminuria is an early marker of chronic renal damage and a risk factor of cardiovascular disease [44], there was likely some misclassification of study groups due to the lack of information concerning the total amount of proteinuria and the presence or absence of microalbuminuria. Third, the estimation of renal function was based on a single measurement of eGFR at the time of presentation to the hospital. However, there may be a possibility that eGFR may have worsened during the follow-up period. Unfortunately, we could not provide follow-up eGFR values because of the limitation of this registry data. Fourth, according to the current guidelines [7], the treatment goal is an LDL-cholesterol level <1.8 mmol/L (<70 mg/dL) or at least 50% reduction in LDL-cholesterol if the baseline LDL-cholesterol level is 1.8–3.5 mmol/L. However, information regarding the follow-up levels of blood LDL-cholesterol was incomplete in our registry data. This is a major shortcoming of this study and may be an important bias. Fifth, because the registry data did not include detailed or complete data on prescription doses, long-term adherence, discontinuation, and drug-related adverse events, we could not provide this information during the follow-up period, which could have caused bias. Sixth, despite multivariable analyses, the variables that were not included in the data registry might have affected the study outcome. Seventh, because statins have a longer duration of use, the 2-year follow-up period in this study was relatively short to estimate the long-term clinical outcomes. Finally, because this retrospective study enrolled patients who underwent PCI between January 2006 and June 2015, this broad timeframe could have affected the clinical outcomes.

Inevitably, because reference numbers 40 and 41 are newly added, original reference numbers 40, 41, and 42 are renumbered as 42, 43, and 44.

Point 3: Did the authors observed side effects of the drug-eluting stents such as thrombosis.

Response 3: We sincerely thank you for reviewer’s valuable question. Yes, we observed definite or probable stent thrombosis (ST) in our study. Both in group A (group A1 vs. A2 vs. A3 vs. A4 = 37/6847 [0.5%] vs. 46/6557 [0.7%] vs. 22/2144 [1.0] vs. 5/507 [1.0%]) and B (9/889 [1.0%] vs. 7/1227 [0.6%] vs. 4/537 [0.7%] vs. 4/537 [0.7%] vs. 1/167 [0.6%], p = 0.702, respectively), the cumulative incidences of ST was very low, and we cannot obtain accurate adjusted HR during the multivariate Cox regression analysis. Therefore, we could not include this variable as an endpoint in our study, inevitably. Therefore, we included this content in the “Limitation” section as follows;

Before,

This study had other limitations. First, there may have been some underreporting and/or missed data and selection bias because this was a nonrandomized study. Second, although micaroalbuminuria is an early marker of chronic renal damage and a risk factor of cardiovascular disease [44], there was likely some misclassification of study groups due to the lack of information concerning the total amount of proteinuria and the presence or absence of microalbuminuria. Third, the estimation of renal function was based on a single measurement of eGFR at the time of presentation to the hospital. However, there may be a possibility that eGFR may have worsened during the follow-up period. Unfortunately, we could not provide follow-up eGFR values because of the limitation of this registry data. Fourth, according to the current guidelines [7], the treatment goal is an LDL-cholesterol level <1.8 mmol/L (<70 mg/dL) or at least 50% reduction in LDL-cholesterol if the baseline LDL-cholesterol level is 1.8–3.5 mmol/L. However, information regarding the follow-up levels of blood LDL-cholesterol was incomplete in our registry data. This is a major shortcoming of this study and may be an important bias. Fifth, because the registry data did not include detailed or complete data on prescription doses, long-term adherence, discontinuation, and drug-related adverse events, we could not provide this information during the follow-up period, which could have caused bias. Sixth, despite multivariable analyses, the variables that were not included in the data registry might have affected the study outcome. Seventh, because statins have a longer duration of use, the 2-year follow-up period in this study was relatively short to estimate the long-term clinical outcomes. Finally, because this retrospective study enrolled patients who underwent PCI between January 2006 and June 2015, this broad timeframe could have affected the clinical outcomes.

After,

This study had other limitations. First, there may have been some underreporting and/or missed data and selection bias because this was a nonrandomized study. Second, although micaroalbuminuria is an early marker of chronic renal damage and a risk factor of cardiovascular disease [44], there was likely some misclassification of study groups due to the lack of information concerning the total amount of proteinuria and the presence or absence of microalbuminuria. Third, the estimation of renal function was based on a single measurement of eGFR at the time of presentation to the hospital. However, there may be a possibility that eGFR may have worsened during the follow-up period. Unfortunately, we could not provide follow-up eGFR values because of the limitation of this registry data. Fourth, according to the current guidelines [7], the treatment goal is an LDL-cholesterol level <1.8 mmol/L (<70 mg/dL) or at least 50% reduction in LDL-cholesterol if the baseline LDL-cholesterol level is 1.8–3.5 mmol/L. However, information regarding the follow-up levels of blood LDL-cholesterol was incomplete in our registry data. This is a major shortcoming of this study and may be an important bias. Fifth, because the registry data did not include detailed or complete data on prescription doses, long-term adherence, discontinuation, and drug-related adverse events, we could not provide this information during the follow-up period, which could have caused bias. Sixth, despite multivariable analyses, the variables that were not included in the data registry might have affected the study outcome. Seventh, because statins have a longer duration of use, the 2-year follow-up period in this study was relatively short to estimate the long-term clinical outcomes. Eighth, because this retrospective study enrolled patients who underwent PCI between January 2006 and June 2015, this broad timeframe could have affected the clinical outcomes. Finally, during a 2-year follow-period, we experienced definite or probable stent thrombosis (ST). Both in group A (group A1 vs. A2 vs. A3 vs. A4 = 37/6847 [0.5%] vs. 46/6557 [0.7%] vs. 22/2144 [1.0] vs. 5/507 [1.0%], p = 0.091) and B (9/889 [1.0%] vs. 7/1227 [0.6%] vs. 4/537 [0.7%] vs. 4/537 [0.7%] vs. 1/167 [0.6%], p = 0.702, respectively), the cumulative incidences of ST were very low. Therefore, although ST is an important major determinant variable in patients with AMI [18], we could not include this variable as an endpoint in our study, inevitably.

Point 4: Please state clearly in the abstract why statin treatment was not effective in reducing mortality in patients with renal disease.

Response 4: We sincerely thank you for reviewer’s valuable comment and recommendation. Regarding words limitations of this journal, we revised “Abstract” as follows according to reviewer’s recommendation.

Before,

Abstracts

We investigated the 2-year efficacy of statin treatment according to baseline renal function in patients with acute myocardial infarction (AMI) not requiring dialysis undergoing newer-generation drug-eluting stent (DES) implantation. A total of 18875 AMI patients were classified into group A (statin users, n = 16055) and group B (statin nonusers, n = 2820). According to the baseline estimated glomerular filtration rate (eGFR; ≥ 90, 60-89, 30-59, and <30mL/min/1.73m2), these two groups were sub-classified into groups A1, A2, A3, and A4 and groups B1, B2, B3, and B4. The major adverse cardiac events (MACE), defined as all-cause death, recurrent MI (re-MI), and any repeat revascularization were evaluated. The MACE (group A1 vs. B1, p = 0.002; group A2 vs. B2, p = 0.007; group A3 vs. B3, p <0.001; group A4 vs. B4, p <0.001), all-cause death (p = 0.006, p <0.001, p <0.001, p <0.001, respectively), and cardiac death (p = 0.004, p <0.001, p <0.001, p <0.001, respectively) rates were significantly higher in statin nonusers than those in statin users. Despite the beneficial effects of statin treatment, the MACE (group A1 vs. A2 vs. A3 vs. A4: 5.2%, 6.4%, 10.1%, and 18.5%, respectively), all-cause mortality (0.9%, 1.8%, 4.6%, and 12.9%, respectively), and cardiac death (0.4%, 1.0%, 2.6%, and 6.8%, respectively) rates were significantly increased as eGFR decreased in group A. In the era of newer-generation DES, although statin treatment was effective in reducing mortality, this beneficial effect was diminished in accordance with the deterioration of baseline renal function.

After,

Abstracts

We investigated the 2-year efficacy of statin treatment according to baseline renal function in patients with acute myocardial infarction (AMI) not requiring dialysis undergoing newer-generation drug-eluting stent (DES) implantation. A total of 18875 AMI patients were classified into group A (statin users, n = 16055) and group B (statin nonusers, n = 2820). According to the baseline estimated glomerular filtration rate (eGFR; ≥ 90, 60-89, 30-59, and <30mL/min/1.73m2), these two groups were sub-classified into groups A1, A2, A3, and A4 and groups B1, B2, B3, and B4. The major adverse cardiac events (MACE), defined as all-cause death, recurrent MI (re-MI), and any repeat revascularization were evaluated. The MACE (group A1 vs. B1, p = 0.002; group A2 vs. B2, p = 0.007; group A3 vs. B3, p <0.001; group A4 vs. B4, p <0.001), all-cause death (p = 0.006, p <0.001, p <0.001, p <0.001, respectively), and cardiac death (p = 0.004, p <0.001, p <0.001, p <0.001, respectively) rates were significantly higher in statin nonusers than those in statin users. Despite the beneficial effects of statin treatment, the MACE (group A1 vs. A2 vs. A3 vs. A4: 5.2%, 6.4%, 10.1%, and 18.5%, respectively), all-cause mortality (0.9%, 1.8%, 4.6%, and 12.9%, respectively), and cardiac death (0.4%, 1.0%, 2.6%, and 6.8%, respectively) rates were significantly increased as eGFR decreased in group A. These results may be related with the peculiar characteristics chronic kidney disease including increased vascular calcification, and traditional or nontraditional cardiovascular risk factors. In the era of newer-generation DES, although statin treatment was effective in reducing mortality, this beneficial effect was diminished in accordance with the deterioration of baseline renal function.

We thank you very much for Reviewer’s efforts in evaluating our submission. We also thank the Reviewer 1 for the helpful and valuable comments, which was very helpful to us to improve our manuscript.

We hope respected Reviewer 1 is healthy and happy

Yong Hoon Kim, M.D., Ph.D.

Reviewer 2 Report

Yong Hoon Kim and colleagues investigated the 2-year efficacy of statin treatment according to baseline renal function  in patients with acute myocardial infarction not requiring dialysis undergoing newer-generation DES implantation. They classified a large (18875) group of AMI patients in  statin users and statin non users and showed that MACE, all-cause death, and cardiac death are significantly higher in statin non users . however, statin  beneficial effect was diminished in accordance with the deterioration of baseline renal function in patients with AMI who underwent successful PCI, improving the knowledge about statin effect on AMI patients.

The study was well designed and clearly presented. It has several limitations that however are already underlined by the same authors in the manuscript.

I think inserting table s2 and s4 in the main manuscript would be useful for the readers

Author Response

Response to Reviewer 2 Comments

Comments and Suggestions for Authors

Yong Hoon Kim and colleagues investigated the 2-year efficacy of statin treatment according to baseline renal function in patients with acute myocardial infarction not requiring dialysis undergoing newer-generation DES implantation. They classified a large (18875) group of AMI patients in  statin users and statin non users and showed that MACE, all-cause death, and cardiac death are significantly higher in statin non users. However, statin  beneficial effect was diminished in accordance with the deterioration of baseline renal function in patients with AMI who underwent successful PCI, improving the knowledge about statin effect on AMI patients.

The study was well designed and clearly presented. It has several limitations that however are already underlined by the same authors in the manuscript.

I think inserting table s2 and s4 in the main manuscript would be useful for the readers

Point 1: I think inserting table s2 and s4 in the main manuscript would be useful for the readers

Response 1: We sincerely thank you for reviewer’s valuable comments and recommendations. According to reviewer’s recommendations, we inserted Table S2, S3, and S4 as Table 2, 3, and 4. Therefore, original Table 2, 3, and 4 are renamed as Table 5, 6, and 7. Moreover, the original Table S5 and S6 are renamed as Table S2 and S3.

Before,

Table legends

Table 1. Baseline characteristics of statin users.

Table 2. Hazard ratios for 2-year major clinical outcomes in statin users.

Table 3. Clinical outcomes between statin users and nonusers at 2 years.

Table 4. Independent predictors for MACE and all-cause death in statin users

After,

Table legends

Table 1. Baseline characteristics of statin users.

Table 2. Baseline characteristics in statin nonusers.

Table 3. Baseline characteristics between statin users and nonusers 1.

Table 4. Baseline characteristics between statin users and nonusers 2.

Table 5. Hazard ratios for the 2-year major clinical outcomes in statin users.

Table 6. Clinical outcomes between statin users and nonusers at 2 years.

Table 7. Independent predictors for MACE and all-cause death in statin users.

Before,

Supplementary Materials: Table S1: Univariate analysis for MACE. Table S2: Baseline characteristics in statin nonusers.            Table S3: Baseline characteristics between statin users and nonusers 1. Table S4: Baseline characteristics between statin users and nonusers 2. Table S5: Hazard ratios for the 2-year major clinical outcomes in statin nonusers. Table S6: Independent predictors for MACE and all-cause death in statin nonusers.

After,

Supplementary Materials: Table S1: Univariate analysis for MACE. Table S2: Hazard ratios for the 2-year major clinical outcomes in statin nonusers. Table S3: Independent predictors for MACE and all-cause death in statin nonusers.

We thank you very much for Reviewer’s efforts in evaluating our submission. We also thank the Reviewer for the helpful and valuable comments, which was very helpful to us to improve our manuscript.

We hope respected Reviewer 2 is healthy and happy

Yong Hoon Kim, M.D., Ph.D.

Round 2

Reviewer 1 Report

The authors answered most of my concerns